# BeliefMapNav: 3D Voxel-Based Belief Map for Zero-Shot Object Navigation

**Zibo Zhou**[1], **Yue Hu**[2], **Lingkai Zhang**[1], **Zonglin Li**[1], **Siheng Chen**[1]*
[1]Shanghai Jiao Tong University
[2]University of Michigan
{zb_zhou, zhanglingkai, channeler_xfya, sihengc}@sjtu.edu.cn
huyu@umich.edu

## Abstract

Zero-shot object navigation (ZSON) allows robots to find target objects in unfamiliar environments using natural language instructions, without relying on pre-built maps or task-specific training. Recent general-purpose models, such as large language models (LLMs) and vision-language models (VLMs), equip agents with semantic reasoning abilities to estimate target object locations in a zero-shot manner. However, these models often greedily select the next goal without maintaining a global understanding of the environment and are fundamentally limited in the spatial reasoning necessary for effective navigation. To overcome these limitations, we propose a novel 3D voxel-based belief map that estimates the target's prior presence distribution within a voxelized 3D space. This approach enables agents to integrate semantic priors from LLMs and visual embeddings with hierarchical spatial structure, alongside real-time observations, to build a comprehensive 3D global posterior belief of the target's location. Building on this 3D voxel map, we introduce BeliefMapNav, an efficient navigation system with two key advantages: i) grounding LLM semantic reasoning within the 3D hierarchical semantics voxel space for precise target position estimation, and ii) integrating sequential path planning to enable efficient global navigation decisions. Experiments on HM3D and HSSD benchmarks show that BeliefMapNav achieves state-of-the-art (SOTA) Success Rate (SR) and Success weighted by Path Length (SPL), with a notable **9.7** SPL improvement over the previous best SR method, validating its effectiveness and efficiency. The source code is publicly available at: https://github.com/ZiboKNOW/BeliefMapNav

## 1 Introduction

Zero-shot object navigation (ZSON) enables robots to locate targets in novel environments through natural language instructions (e.g., "find the red sofa"), eliminating reliance on pre-mapped scenes or object-specific training [1, 2, 3, 4, 5]. In domestic settings, ZSON supports assistive tasks such as retrieving user-specified objects [6]. In industrial inspection, ZSON enables autonomous localization of malfunctioning components (e.g., detecting a leaking pipe) within complex facilities. In warehouse operations, ZSON enhances robotic picking and inventory management by allowing flexible retrieval of objects without pre-built maps. These real-world applications highlight the necessity of robust zero-shot navigation for scalable, adaptable robot deployment across diverse domains.

To enable ZSON, prior works have progressed along two main directions. The first approach constructs bird's-eye view (BEV) value maps [7, 8, 9] by leveraging pixel-level semantic cues to

---

*Corresponding author.

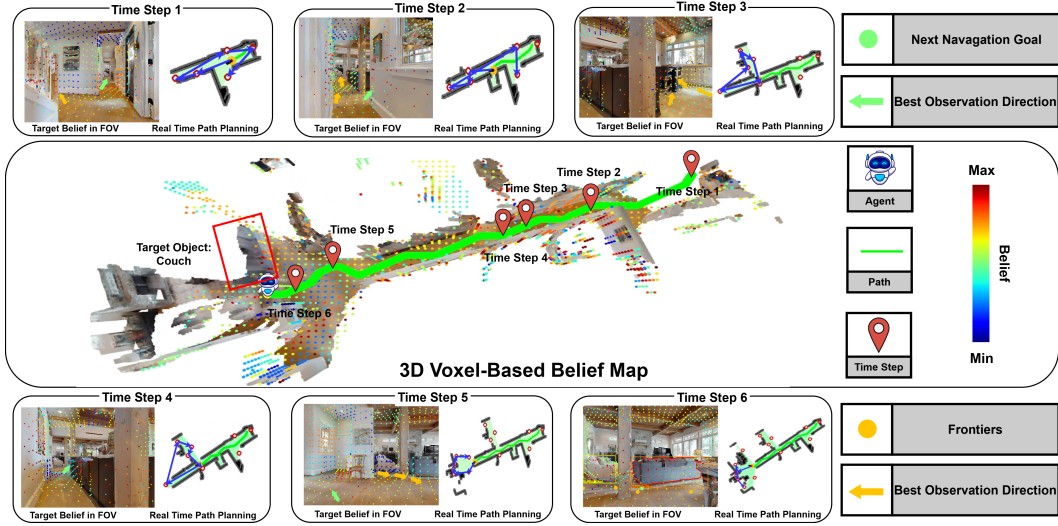

Figure 1: The search process: BeliefMapNav plans frontier paths by minimizing the expected search distance based on the 3D voxel-based belief map, ensuring efficient and stable exploration.

provide dense estimations of target locations in the BEV space. While dense BEV representations provide object position estimation at a fine-grained level, the actual numerical distinctions between positions remain ambiguous with limited discriminative semantics. This insufficient differentiation in positional values, combined with the lack of high-level semantic reasoning, ultimately leads to compromised accuracy in the target location prediction. More recently, a second approach has emerged that employs large language models (LLMs) or vision-language models (VLMs) to reason about the target locations [10, 11, 12, 13, 14]. However, both LLMs and VLMs face limitations in spatial understanding and reasoning [15], which significantly affect target location prediction accuracy. Additionally, VLMs are limited by their inability to effectively extract the most relevant and fine-grained semantic information from image observations, as well as by the lack of spatial information, resulting in imprecise target position predictions [10, 11, 12, 13, 14]. For LLMs, while leveraging spatial information, reliance on semantic content from environment language descriptions leads to significant loss of information and reduced prediction precision [16, 17, 12, 18]. Consequently, LLMs and VLMs both struggle with reliable and accurate target location inference. Furthermore, existing methods generally rely on greedy navigation strategies [12, 8], which cause frequent back-and-forth movements, significantly hindering search efficiency. Together, in existing works, the lack of semantic cues and spatial reasoning leads to inaccurate and imprecise target object position estimation. Meanwhile, the absence of efficiency optimization in the search behavior hinders robust searching for diverse, open-set targets in unbounded real-world 3D spaces.

To enable more precise and accurate predictions of the target object's location within 3D space, we propose a novel 3D voxel-based belief map that considers rich hierarchical spatial semantic cues and LLM-generated target-adaptive semantic cues to reason about prior belief of target presence in dense 3D voxel space. This structured representation enables spatially fine-grained estimation across the 3D voxel space, facilitating more precise and generalizable localization of target objects in complex, unbounded environments. Moreover, this fine-grained and accurate representation enables efficient guidance for high-degree-of-freedom mobile agents, facilitating precise, task-relevant, language-driven search within localized areas and enhancing the robustness of manipulation tasks.

To further enhance search efficiency, we introduce BeliefMapNav, an efficient zero-shot object navigation system based on path sequence optimization over the belief map. The system is composed of three tightly integrated modules and builds upon the belief map framework to enable efficient, goal-directed exploration. 1) The **3D voxel-based belief mapping module** encodes prior beliefs of object presence in 3D space by integrating hierarchical spatial semantics with commonsense knowledge from an LLM. 2) The **frontier observation belief estimation module** combines the belief map with a visibility map, which encodes real-time observation feedback likelihood, to produce posterior beliefs of the target object's position. Then, the module estimates the posterior observation belief of detecting the target in each frontier's field of view (FOV). 3) The **observation belief-based planning module** selects the next navigation goal by minimizing the expected distance cost based on the posterior observation beliefs to find the target. By explicitly modeling uncertainty and updating

spatial beliefs dynamically, BeliefMapNav enables more efficient, goal-directed exploration than methods using static priors or reactive policies.

The contributions of our method are mainly summarized as follows: 1) We propose BeliefMapNav, an efficient zero-shot object navigation system that accurately predicts target location through fine-grained belief estimation in a 3D voxel-based belief map and real-time feedback, enabling belief-driven sequential planning for efficient navigation. 2) At the core of our system is a 3D voxel-based belief map that integrates hierarchical spatial-visual features with LLM-derived commonsense, enabling accurate and fine-grained prior estimation of the target's location. Based on the prior estimation, we design a planner that optimizes navigation by minimizing the expected path distance cost for efficient navigation. 3) The proposed system achieves SOTA performance on the HM3D [19] and HSSD [20] benchmarks, surpassing all the previous zero-shot methods by 3.4% in SR and 9.7 in SPL on HM3D, 14.2% in SR and 7.2 in SPL on HSSD, thereby demonstrating the overall effectiveness and efficiency of our approach.

## 2   Related Works

**Object Navigation** Object navigation refers to the task of guiding a robot to search given target objects in an unknown environment. It can be divided into two categories: i) training-required methods, such as reinforcement learning (RL) and imitation learning, which require extensive training on task-specific data [21, 22, 23, 24, 25], and ii) zero-shot methods, which leverage pre-trained models, such as VLMs or LLMs, to perform navigation without additional training [14, 3, 9, 8, 12]. Training-based methods typically require large amounts of data and have difficulty generalizing due to limited environmental diversity [19, 26], while zero-shot methods offer flexibility and adaptability to novel environments, but are constrained by the spatial reasoning limitations of LLMs and VLMs [15]. Our method follows a zero-shot approach. We leverage LLMs and rich hierarchical spatial semantics to provide accurate and fine-grained estimations of the target's location, while employing probability-based optimization algorithms to ensure the efficiency of the search path.

**Semantic Mapping for Object Navigation.** Semantic mapping is important in object navigation as it provides spatially structured representations that guide the robot in locating targets. Traditional methods, such as category-based approaches [14, 27, 17] and scene graph-based methods [28, 16, 18, 29, 30, 31], often rely on predefined categories or topological graphs, leading to semantic information loss and mapping errors due to detection failures [32, 33]. Although value map-based methods [8, 7, 9] aim for non-vocabulary representations, existing methods struggle to align spatial and semantic scales, resulting in incomplete or misaligned spatial semantics under varying scene extents. As a result, the generated maps lack the precision needed to accurately localize target objects. In contrast, our method constructs a multi-level, spatially-aligned semantic map that supports accurate target object localization estimation.

## 3   Method

In this section, we first define the object navigation task and then present the BeliefMapNav system, which consists of three main modules: 3D voxel-based belief mapping, frontier observation belief estimation, and observation belief-based planning, as shown in Fig. 2.

### 3.1   Task definition

We define the ZSON task, where an agent is required to locate a specified target object in an unknown environment without task-specific training, pre-built maps, or a fixed vocabulary. The target category $c$ is specified in free-form text. At each timestep $t$, the agent receives RGB-D observations $I_t = (I_t^{\mathrm{rgb}}, I_t^{\mathrm{depth}})$, where $I_t^{\mathrm{rgb}} \in \mathbb{R}^{H \times W \times 3}$ and $I_t^{\mathrm{depth}} \in \mathbb{R}^{H \times W}$, and its pose $s_t = (x_t, r_t)$, with $x_t \in \mathbb{R}^3$ and $r_t \in \mathrm{SO}(3)$, from odometry. The action space $\mathcal{A}$ includes: MOVE FORWARD (0.25 m), TURN LEFT/RIGHT (30°), LOOK UP/DOWN (30°), and STOP. The task is successful if the agent issues a STOP within 0.1 m of the target object within 500 steps. At each timestep, the system takes as input the current RGB-D observation $I_t$, the agent's pose $s_t$, and the text-specified target $c$, and outputs a navigation action $a_t \in \mathcal{A}$ from the discrete action set.

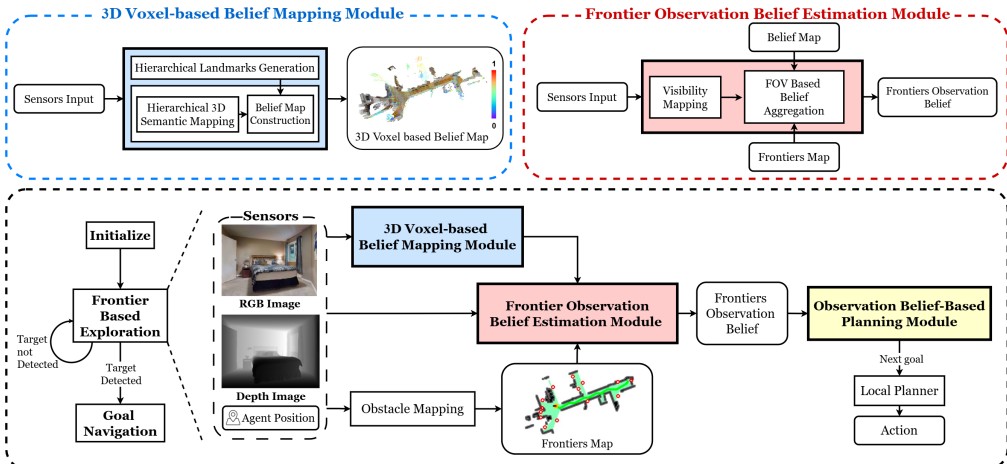

Figure 2: BeliefMapNav pipeline: The agent initializes with a 360° rotation. During exploration, the 3D voxel-based belief mapping module fuses sensor input, the 3D hierarchical semantic map, and landmarks to create a belief map. The frontier observation belief estimation module computes frontier observation belief from the belief, frontiers, and visibility maps via FOV-based aggregation. The observation belief-based planning module selects the next goal based on this belief and outputs navigation actions. Upon detecting the target, the agent navigates to it.

## 3.2 System overview

BeliefMapNav is a novel 3D voxel-based zero-shot open-vocabulary object navigation system; see the overview in Fig. 2. BeliefMapNav has three key modules: 1) 3D voxel-based belief mapping module in Sec. 3.3, which encodes a prior belief of the target's presence by combining hierarchical spatial semantics with LLM commonsense in a belief map; 2) frontier observation belief estimation module in Sec. 3.4, which fuses the prior belief map with real-time observation likelihood to estimate the posterior belief of detecting the target in each frontier's FOV; 3) observation belief-based planning module in Sec. 3.5, which selects the next navigation point by optimizing expected distance cost to detect the object.

## 3.3 3D voxel-based belief mapping

3D voxel-based belief mapping aims to represent the spatial prior belief of the target object's presence in a 3D voxel grid. The key intuition is that maintaining a fine-grained representation enables more spatially detailed and accurate prediction of the target's location. To achieve this, we construct a 3D voxel map where each voxel stores the belief of the target existing within its spatial region:

$$\mathcal{B} = \left\{ (u, b_u) \mid u \in \mathbb{Z}^3 \right\}$$

where $\mathcal{B}$ denotes the belief map, $u \in \mathbb{Z}^3$ represents the discrete spatial coordinate of a voxel in 3D space and $b_u \in \mathbb{R}$ represents the estimated prior belief that the target object exists within voxel $u$. Unlike previous methods [7, 8], the 3D voxel-based belief map leverages hierarchical language and spatial semantics to provide more precise and nuanced estimation of the target belief in each voxel of 3D space. It involves three steps as shown in Fig. 3: 1) constructing a 3D hierarchical semantic voxel map based on visual observations in Sec. 3.3.1; 2) using an LLM to infer hierarchical landmarks with corresponding relevance scores in Sec. 3.3.2; and 3) mapping the inferred landmarks and relevance scores into the hierarchical semantic voxel map to form the belief map in Sec. 3.3.3.

### 3.3.1 3D Hierarchical semantic mapping

The 3D hierarchical semantic voxel map $\mathcal{M}_c$ represents the environment across three levels, $\mathcal{L}_s = \{\text{scene}, \text{region}, \text{object}\}$, with progressively finer semantics. This structure enables reasoning across spatial scales, from coarse layouts to fine object details, improving the accuracy of target object position estimation. Formally: $\mathcal{M}_c = \left\{ \left(u, \{\hat{v}_u^{l_s}, \hat{s}_u^{l_s}\}\right) \mid u \in \mathbb{Z}^3, l_s \in \mathcal{L}_s \right\}$. Here, for each level $l_s \in \mathcal{L}_s$, voxel $u$ stores an image CLIP [34] feature vector $\hat{v}_u^{l_s} \in \mathbb{R}^d$ and a feature confidence score

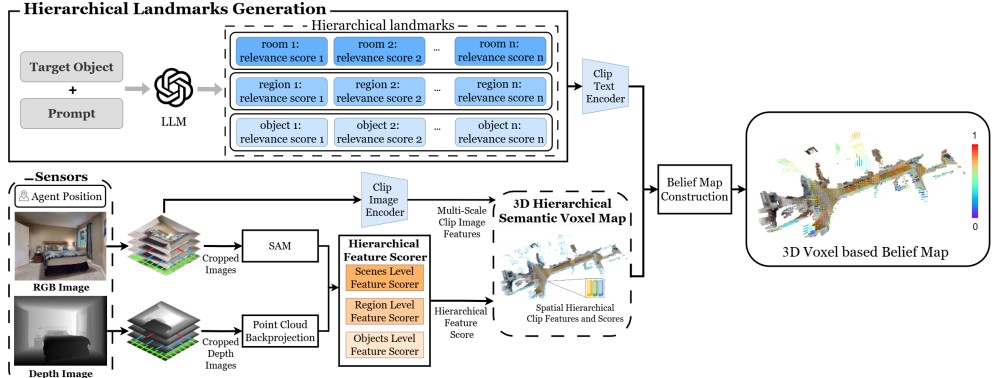

Figure 3: The pipeline begins by passing the target and prompt to the LLM to generate hierarchical landmarks with relevance scores. Meanwhile, RGB and depth images are cropped into multi-scale patches and processed via SAM and point cloud back-projection. Multi-scale CLIP image features are extracted, and the top features selected by hierarchical feature scorers update the hierarchical 3D semantic map. Finally, landmarks encoded by text CLIP and the semantic map are combined via the belief map construction to update the 3D voxel-based belief map.

$\hat{s}_u^{l_s} \in \mathbb{R}$, which quantifies the reliability of the semantic feature at each level and guides the selection of the most informative features for belief map updating.

The module operates in three stages: 1) **Multi-scale feature extraction**: Extract image CLIP features from multi-scale RGB images and spatial information from the depth images. 2) **Hierarchical feature scoring**: Assign confidence scores to features at different image scales, reflecting their relevance to specific semantic levels $\mathcal{L}_s$. These scores enable adaptive scale selection for each semantic level. 3) **Adaptive hierarchical feature selection**: At each hierarchical level, features from the most confident scale are selected and back-projected to update the 3D voxel map $\mathcal{M}_c$.

**Multi-scale feature extraction**: To better capture both global context and local details, the observed RGB image $I^{rgb} \in \mathbb{R}^{H \times W \times 3}$ is divided into equal-sized patches at multiple scales. At scale $k$, the image is partitioned into $2^{(k-1)} \times 2^{(k-1)}$ patches, with each patch denoted as $P_{h,w}^k$. We use CLIP [34] to extract visual features $v_{h,w}^k$ for each patch and each patch $P_{h,w}^k$ is processed by the Segment Anything Model (SAM) [35] to estimate the number of semantic instances $n_{h,w}^k$. In parallel, the corresponding depth image is divided in the same way. We back-project depth values of patches into 3D space to form a point cloud and compute two geometric properties: the volume $V_{h,w}^k$ and point density $\rho_{h,w}^k$. The detailed extraction process is in Appendix A.1.

**Hierarchical feature scoring**: To select features for different spatial semantic levels, we design hierarchical feature scorers for $L_s$, which assign confidence scores to each image patch. A higher score indicates a better alignment of the feature with the corresponding semantic level. The hierarchical feature scorers are defined as: **(1) Scene Scorer:** This scorer favors patches covering larger spatial extents with more semantic instances, score defined as $S_{h,w}^{\text{scene},k} = w_1 \cdot V_{h,w}^k + w_2 \cdot n_{h,w}^k$. **(2) Region Scorer:** This scorer favors patches with more densely packed instances and concentrated point clouds, score defined as $S_{h,w}^{\text{region},k} = w_3 \cdot \left( \frac{n_{h,w}^k}{V_{h,w}^k} \right) + w_4 \cdot \rho_{h,w}^k$. **(3) Object Scorer:** This scorer favors patches with a high average point density per instance, score defined as $S_{h,w}^{\text{object},k} = \frac{\rho_{h,w}^k}{n_{h,w}^k}$

**Adaptive hierarchical feature selection**: Each image pixel position has $k$ scale candidate features, and in each pixel position, we select the image CLIP features for hierarchical semantic levels from multi-scale patches that achieve the highest score under the corresponding semantic-level feature scorer. After scoring, pixels are back-projected into the 3D semantic map, where for each semantic level in every voxel, only the feature with the highest confidence score is retained in the map. The detailed selection method is provided in the Appendix A.2.

### 3.3.2 Hierarchical landmarks generation

Landmarks are semantic cues inferred by a language model from the target object description, indicating where the object is likely to appear. In our method, we focus on three levels of landmarks $\mathcal{L}_t = \{\text{room}, \text{region}, \text{object}\}$ to assist in locating the target object. To extract these landmarks, we prompt an LLM (GPT-4o [36]) with the target object description, asking it to generate two outputs: (1) a set of landmark strings $\mathcal{R} = [\mathcal{R}^{\text{room}}, \mathcal{R}^{\text{region}}, \mathcal{R}^{\text{object}}]$, where $\mathcal{R}^{l_t} = [r_1^{l_t}, r_2^{l_t}, \ldots, r_{n_{l_t}}^{l_t}]$ denotes the landmarks at level $l_t \in \mathcal{L}_t$; and (2) the corresponding relevance scores $\alpha = [\alpha^{\text{room}}, \alpha^{\text{region}}, \alpha^{\text{object}}]$, where $\alpha^{l_t} = [\alpha_1^{l_t}, \alpha_2^{l_t}, \ldots, \alpha_{n_{l_t}}^{l_t}]$. Where $n_{l_t}$ is the number of landmarks in level $l_t$. Each score $\alpha_i^{l_t}$ indicates the likelihood that the target object appears near the corresponding landmark $r_i^{l_t}$. Details about the prompt design and generation process are provided in the Appendix A.3.

### 3.3.3 Belief map construction

After obtaining hierarchical textual landmarks with associated relevance scores, we project both the landmarks and the target object name into the 3D hierarchical semantic voxel map to generate a 3D voxel-based belief map, which represents a prior belief over the target object's presence in space. Each landmark $r_i^{l_t} \in \mathcal{R}^{l_t}$ and the target object name $r_{\text{target}}$ are encoded using the CLIP text encoder, resulting in embeddings $\mathcal{E}_i^{l_t}$ for landmarks and $\mathcal{E}_{\text{target}}$ for the target object. For each of these textual inputs, we compute the maximum cosine similarity scores with stored spatial semantic features at corresponding levels: $p_{u,i}^{l_t} = \max_{l_s} \text{cosine}(\mathcal{E}_i^{l_t}, \hat{v}_u^{l_s})$ and $p_{u,\text{target}} = \max_{l_s} \text{cosine}(\mathcal{E}_{\text{target}}, \hat{v}_u^{l_s})$. Each similarity score is weighted by its associated relevance score $\alpha_i^{l_t}$ for landmarks and $\alpha_{\text{target}} = 1$ for the target object. The final belief score at voxel $u$ is computed as: $b_u = \sum_{l_t \in \mathcal{L}_t} \sum_{i=1}^{n_{l_t}} \alpha_i^{l_t} \cdot p_{u,i}^{l_t} + p_{u,\text{target}}$. The values $b_u$ in the 3D voxel-based belief map represent the prior belief of the target object's presence in each voxel.

## 3.4 Frontier observation belief estimation

The 3D voxel-based belief map serves as the prior, through which the frontier observation belief module dynamically integrates visibility map likelihoods to calculate the belief of detecting the target object within each frontier's FOV.

### 3.4.1 Visibility map

To capture the impact of real-time detection feedback on the target belief distribution, we introduce a visibility map. The visibility map is inspired by [37] and is defined as $\mathcal{P}^v = \{(u, \hat{p}_u^v) \mid u \in \mathbb{Z}^3, \hat{p}_u^v \in [0,1]\}$, $\hat{p}_u^v$ is the likelihood of detecting the target at voxel $u$ based on visual observations during the search. The key intuition is that if a voxel $u$ has high detection confidence in the FOV but no detection, the belief that the target object exists in that region is very low ($\hat{p}_u^v \to 0$). By estimating the object absence likelihood, the visibility map can refine prior beliefs to prevent revisiting areas that have been well observed. This correction improves navigation efficiency. The detection confidence is calculated in a way that it decreases near image boundaries and at longer distances from the camera pose. Details of the construction of the visibility map are provided in the Appendix A.4.

### 3.4.2 FOV-based belief aggregation

After constructing the visibility map $\mathcal{P}^v$ and belief map $\mathcal{B}$, we fuse them to obtain the **posterior belief map** $\mathcal{B}^{\text{post}} = \{(u, \hat{b}_u^{\text{post}}) \mid u \in \mathbb{Z}^3, \hat{b}_u^{\text{post}} = \hat{p}_u^v \cdot b_u\}$. This fusion enables more dynamic and accurate estimation of the belief of detecting the target from each frontier's FOV by combining spatial priors belief map with the observation feedback visibility map. It improves search efficiency and reduces exploration of well-observed regions. For each candidate frontier position $x_{f_i}$, we evaluate four viewing directions $\theta \in \{0°, 90°, 180°, 270°\}$. For each $\theta$, we perform **ray casting** from $x_{f_i}$ to identify voxels within the FOV, separating them into $\mathcal{O}_{\text{map}}^\theta(x_{f_i})$ (voxels in the belief map) and $\mathcal{O}_{\text{unk}}^\theta(x_{f_i})$ (voxels out of belief map). The observation belief for direction $\theta$ is computed as: $P_{\text{obs}}^\theta(x_{f_i}) = \sum_{u \in \mathcal{O}_{\text{map}}^\theta(x_{f_i})} \hat{b}_u^{\text{post}} + |\mathcal{O}_{\text{unk}}^\theta(x_{f_i})| \cdot w_{\text{unobserved}}$. Where $|\mathcal{O}_{\text{unk}}^\theta(x_{f_i})|$ is the number of

voxels not in map and $w_{\text{unobserved}}$ is a constant weight. The final observation belief at $x_{f_i}$ is defined as: $P_{\text{obs}}(x_{f_i}) = \max_\theta P_{\text{obs}}^\theta(x_{f_i})$. Aggregating over the FOV allows us to account for obscure vision and the agent's limited FOV, leading to a more accurate estimation of the likelihood of observing the target in each frontier's FOV.

## 3.5 Observation belief-based planning module

Using the estimated observation belief at each frontier based on the posterior belief map, we prioritize frontiers to minimize expected search distance for a more stable and efficient target search. Unlike previous greedy approaches that myopically prioritize immediate gain, our distance-based optimization ensures smoother early-stage movement under uncertain belief distributions and gradually converges to the optimal path as the belief becomes more reliable, resulting in more efficient exploration. The optimal exploration strategy seeks a permutation of frontier visiting sequence $\pi = [f_{\pi_1}, f_{\pi_2}, \ldots, f_{\pi_n}]$ that minimizes the expected search cost, where $f_{\pi_i} \in \{x_{f_1}, \ldots, x_{f_n}\}$ denotes the $i$-th frontier position. The objective is formulated as:

$$\pi^* = \operatorname*{argmin}_{\pi \in S_n} \sum_{i=1}^{n} \left( \sum_{k=1}^{i} d_{A^*}(f_{\pi_{k-1}}, f_{\pi_k}) \right) P_{\text{obs}}(f_{\pi_i})$$

where $S_n$ denotes the permutation group over $n$ frontier points, $f_{\pi_k}$ represents the $k$-th visited frontier in permutation $\pi$ (with the initial point $f_{\pi_0} \equiv x_0$ defined as the agent's current position), $d_{A^*}(f_{\pi_{k-1}}, f_{\pi_k})$ denotes the path distance between adjacent frontiers computed via the A* [38] algorithm, $\sum_{k=1}^{i} d_{A^*}(\cdot)$ calculates the cumulative path cost to the $i$-th frontier, and $P_{\text{obs}}(f_{\pi_i})$ is the observation belief of frontier $f_{\pi_i}$.

The proposed objective improves search efficiency by minimizing exploration cost with A*-optimized paths and prioritizing high-belief frontiers via observation-weighted costs. Combining geometric path costs and belief weights, it reduces noise impact on navigation stability. Integrating shortest-path guarantees with probabilistic reasoning, it enables dynamic, real-time replanning, adapting to evolving beliefs for flexible, efficient navigation. At the same time, the precise and detailed belief map provides a solid foundation for effective optimization. We solve this optimization via GPU-accelerated simulated annealing [39]. The detailed optimization process is in the Appendix A.5. Before each action, the agent selects the first frontier in the optimized sequence $\pi^*$ as the next navigation target and replans at every step with the updated belief map. We adopt the local point navigation planner from VLFM [7] to generate actions toward the given goal. An open-vocabulary detector [40] and GPT-4o [36] verify detected objects; if confirmed, the system localizes the target using the bounding box, SAM [35], and depth, then sets it as the final navigation goal.

## 4 Experimental Results

In this section, we outline datasets and key implementation details, then compare BeliefMapNav's performance against SOTA baselines on HM3D [19], MP3D [26], and HSSD [20]. Ablation studies assess each component's contribution. Qualitative analysis visualizes maps indicating target presence probability. Search process visualizations highlighting our approach are in Appendix A.9. Baseline summaries and HM3D failure analyses appear in Appendix A.6 and A.7, respectively.

### 4.1 Benchmarks and Implementation details

**Dataset:** We evaluate our method on three standard benchmarks: HM3D [19], MP3D [26] and HSSD [20]. HM3D, the official dataset of the Habitat 2022 ObjectNav Challenge, includes 2,000 validation episodes across 20 environments and 6 object categories. MP3D, a large-scale indoor 3D scene dataset, is commonly used in Habitat-based ObjectNav evaluations. We conduct experiments on its validation set, consisting of 11 environments, 21 object categories, and 2,195 object-goal navigation episodes. HSSD, a synthetic dataset with scenes based on real house layouts, contains 40 validation scenes, 1,248 navigation episodes, and 6 object categories.

**Evaluation Metrics:** We use two standard metrics: Success Rate (SR) and Success weighted by Path Length (SPL). SR measures the proportion of episodes where the agent reaches the target within a

Table 1: Zero-shot object navigation results on MP3D, HM3D and HSSD. We compare the SR and SPL of state-of-the-art methods in different settings.

| Method | Unsupervised | Zero-shot | HM3D | | MP3D | | HSSD | |
|---|---|---|---|---|---|---|---|---|
| | | | SR↑ | SPL↑ | SR↑ | SPL↑ | SR↑ | SPL↑ |
| Habitat-Web [41] | ✗ | ✗ | 41.5 | 16.0 | 31.6 | 8.5 | - | - |
| OVRL [42] | ✗ | ✗ | - | - | 28.6 | 7.4 | - | - |
| ProcTHOR [43] | ✗ | ✗ | 54.4 | 31.8 | - | - | - | - |
| SGM [44] | ✗ | ✗ | 60.2 | 30.8 | 37.7 | 14.7 | - | - |
| ZSON [24] | ✗ | ✓ | 25.5 | 12.6 | 15.3 | 4.8 | - | - |
| PSL [45] | ✗ | ✓ | 42.4 | 19.2 | 18.9 | 6.4 | - | - |
| PixNav [11] | ✗ | ✓ | 37.9 | 20.5 | - | - | - | - |
| VLFM [7] | ✓ | ✓ | 52.5 | 30.4 | 36.4 | 17.5 | - | - |
| ESC [3] | ✓ | ✓ | 39.2 | 22.3 | 28.7 | 14.2 | 38.1 | 22.2 |
| CoWs [13] | ✓ | ✓ | - | - | 9.2 | 4.9 | - | - |
| L3MVN [14] | ✓ | ✓ | 50.4 | 23.1 | 34.9 | 14.5 | 41.2 | 22.5 |
| ImagineNav [46] | ✓ | ✓ | 53.0 | 23.8 | - | - | 51.0 | 24.9 |
| VoroNav [28] | ✓ | ✓ | 42.0 | 26.0 | - | - | 41.0 | 23.2 |
| GAMap [8] | ✓ | ✓ | 53.1 | 26.0 | - | - | - | - |
| OpenFMNav [47] | ✓ | ✓ | 52.5 | 24.1 | 37.2 | 15.7 | - | - |
| SG-Nav [12] | ✓ | ✓ | 54.0 | 24.9 | 40.2 | 16.0 | - | - |
| UniGoal [48] | ✓ | ✓ | 54.5 | 25.1 | **41.0** | 16.4 | - | - |
| InstructNav [9] | ✓ | ✓ | 58.0 | 20.9 | - | - | - | - |
| **BeliefMapNav** | ✓ | ✓ | **61.4** | **30.6** | 37.3 | **17.6** | **65.2** | **32.1** |

preset distance. SPL evaluates path efficiency by considering both success and trajectory optimality: if successful, $\text{SPL} = \frac{\text{Optimal path length}}{\text{path length}}$, otherwise $\text{SPL} = 0$. Higher values indicate better performance.

**Implementation details**: We limit navigation to 500 steps, defining success as stopping within 0.1 m of the target. Each step moves the agent 0.25 m forward or rotates it by 30°. The RGB-D camera, mounted 0.88 m high, captures 640 × 480 images. The 3D voxel map has 45,000 voxels at 0.25 m resolution. We set $w_{\text{unobserved}} = 0.01$ (Sec. 3.4.2). CLIP-ViT-B-32 encodes visual/text features with image crop scale $k = 3$. GPT-4o generates three landmarks per level (nine total). Hierarchical scorer weights are $w_1 = 0.05$, $w_2 = 0.1$, $w_3 = 2$, $w_4 = 0.01$. The system runs on a single RTX 4090 (24 GB VRAM) and uses approximately 13 GB of VRAM. Local planner parameters vary slightly by dataset, while the exploration module remains unchanged.

## 4.2 Comparison with SOTA methods

In this section, we compare our proposed BeliefMapNav with SOTA object navigation approaches in different settings, including unsupervised, supervised, and zero-shot methods on the MP3D [26], HM3D [19], and HSSD [20] benchmarks. As shown in Table 1, our method outperforms all existing zero-shot baselines except the SR of UniGoal and SG-Nav, achieving significant improvements across multiple benchmarks. Specifically, on **HM3D**, we observe a gain of +3.4% in SR and +0.2 in SPL. On **MP3D**, we achieve +0.1 in SPL. Finally, on **HSSD**, our method delivers remarkable gains of +14.2% in SR and +7.2 in SPL. These results highlight the effectiveness of our approach in enhancing both SR and SPL across diverse datasets. While our SR on the MP3D dataset is marginally lower than that of UniGoal and SG-Nav, this is primarily due to their utilization of an object verification mechanism. This feature specifically targets the reduction of detector false positives (FP) caused by poor mesh quality inherent in MP3D. Crucially, while such object verification is an important technique, it remains orthogonal to our main contribution, which focuses squarely on efficient target-oriented exploration. This distinction is further evidenced by our performance on the high-quality HM3D dataset, where our method significantly outperforms UniGoal in both SR (+6.9%) and SPL (+5.5).

On the HM3D dataset, our method improves SPL by 9.7 compared to the zero-shot method Instruct-Nav [9], which achieves the highest SR. While InstructNav prioritizes SR with a dense search strategy, our approach maintains high success rates and boosts search efficiency by generating more accurate

target position estimates and optimizing the search path with a distance cost-aware planner. On the MP3D benchmark, improvements are less pronounced due to two factors: first, the lower data quality of MP3D, which makes target recognition more challenging. Second, there are a lot of mesh "holes" in MP3D, which allow the agent to see through obstacles, causing it to mistakenly prioritize these holes as targets, leading to navigation failures when it gets stuck near non-traversable areas. However, on the HSSD dataset, performance significantly improves because the synthetic scenes avoid the issues present in MP3D and HM3D. Across all datasets, the performance limitations of the local planner in [7] lead to significant degradation, especially in narrow areas.

## 4.3  Ablation study

To evaluate the effectiveness of each module in our system, we conduct an ablation study on 400 randomly sampled episodes from the HM3D validation set, using a fixed random seed. The ablation study of the effectiveness of LLMs and image scale k are in Appendix A.8.

Table 2: Impact of the planner and visibility map.

| Method | SR↑ | SPL↑ |
|---|---|---|
| BeliefMapNav w/o Planner | 56.0 | 29.3 |
| BeliefMapNav w/o Visibility Map | 57.2 | 28.0 |
| BeliefMapNav | **62.5** | **31.6** |

Table 3: Impact of vision-language encoders.

| Encoder | SR↑ | SPL↑ |
|---|---|---|
| BLIP [49] | 59.3 | 31.0 |
| BLIP2 [50] | 62.0 | 31.1 |
| CLIP [34] | **62.5** | **31.6** |

**Effectiveness of visibility map and belief-based planning**: In Table 2, we compare the effectiveness of the Visibility Map and Belief-based planning against basic exploration. Without the Visibility Map, relying on spatial priors alone leads to a $5.3\%$ ↓ drop in SR and $3.6$ ↓ in SPL, as the agent revisits previously observed regions. Without the planner, navigation is based solely on the highest posterior belief, resulting in a $6.5\%$ ↓ drop in SR and $2.3$ ↓ in SPL due to frequent navigation goal switching and inefficient back-and-forth movement.

**Effectiveness of different levels of hierarchical 3D semantics:** As shown in Table 5, we evaluate the impact of different levels of the Hierarchical 3D Semantic Map on performance, comparing four settings: no semantics (random walk), scene-level only, scene + region levels, and the full hierarchy with object-level semantics. Results indicate that incorporating more semantic levels generally improves SR. However, omitting object-level semantics enhances efficiency (32.0), as fine-grained searches with object-level cues increase success rates but often result in slower, localized exploration, leading to longer paths and slightly reduced efficiency.

Table 4: Impact of different levels of landmarks.

| Landmarks | SR↑ | SPL↑ |
|---|---|---|
| w/o | 60.0 | 30.9 |
| Room | 61.0 | 31.1 |
| Room+Region | 61.5 | 31.2 |
| Room + Region + Object | **62.5** | **31.6** |

Table 5: Impact of different semantic levels.

| Semantics | SR↑ | SPL↑ |
|---|---|---|
| Random Walking | 21.5 | 10.8 |
| Scene | 59.0 | 30.4 |
| Scene + Region | 61.5 | **32.0** |
| Scene + Region + Object | **62.5** | 31.6 |

**Effectiveness of different vision-language encoders**: as shown in Table 3, CLIP- and BLIP-2-based systems achieve comparable performance (SR: 62.5 vs. 62.0; SPL: 31.6 vs. 31.1), both outperforming BLIP. Prior work [51] similarly shows that BLIP-2 slightly surpasses CLIP in zero-shot text-to-image retrieval accuracy, while both are significantly better than BLIP. However, CLIP demonstrates stronger generalization to out-of-distribution data and supports efficient inference via independent encoders and pre-computed features. These properties make CLIP a more robust and practical encoder choice for retrieval-based navigation tasks.

**Effectiveness of hierarchical landmarks:** As shown in Table 4, without landmarks, we retrieve directly using the object name in the hierarchical 3D semantic map. With incremental landmark introduction, we observe a gradual improvement in SR (60 to 62.5) and SPL (30.9 to 31.6). However, this improvement is less significant than the gain from incorporating spatial semantics at different hierarchical levels in Table 5, as object names already show inherent relevance to scene, region, and object semantics. In this context, hierarchical landmarks mainly reinforce these semantic associations.

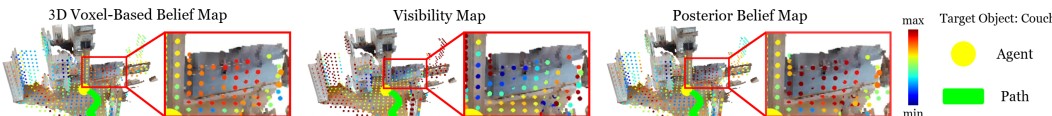

Figure 4: Visualization of the prior belief map, visibility map, and the posterior belief map, with an enlarged section highlighting the target object.

## 4.4 Qualitative Analysis

Fig. 4 shows the 3D voxel-based belief map, visibility map, and posterior belief map, and the complete visualization is in Fig. 12 of Appendix. The posterior belief map assigns high belief to the target object's (couch) location. While the visibility map indicates low likelihood, the prior belief map, guided by vision-and-language cues, strongly suggests the target's presence, effectively guiding the agent's search (Fig. 1). Additional visualizations are provided in Appendix A.9.

## 5 Conclusion

In this paper, we have proposed BeliefMapNav, a zero-shot object navigation system that integrates hierarchical spatial semantics, commonsense reasoning from LLMs, and real-time feedback through a 3D voxel-based belief representation. Compared to prior approaches that rely on planar value maps or local greedy goal selection, BeliefMapNav maintains a global 3D belief posterior, performs visibility-aware updates, and plans over informative frontiers, improving robustness to environmental complexity, reducing backtracking, and enabling more efficient exploration. Experiments on three benchmarks show that it outperforms previous methods. Ablation studies highlight the effectiveness of our belief map and belief-based planning for efficient exploration, emphasizing the system's effectiveness and efficiency.

**Limitation and future work.** We validate the effectiveness of the 3D voxel-based belief map solely on object navigation tasks. This high-resolution 3D map can be further extended to enable robot interaction for mobile manipulation tasks, with future work focusing on real-world implementation.

## Acknowledgements

The authors gratefully acknowledge the computational support provided by the CMIC of Shanghai Jiao Tong University.

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

# A Appendix

## A.1 Multi-scale feature extraction

To better capture both global context and local details, the observed images are divided into equal-sized patches at different scales. Specifically, given an RGB observation image $I^{rgb}$ of height $H$ and width $W$, at image scale $k$, the image is divided into $2^{(k-1)} \times 2^{(k-1)}$ equally-sized patches, resulting in a total of $4^{(k-1)}$ patches. Each patch at level $k$ has image size: $\frac{H}{2^{(k-1)}} \times \frac{W}{2^{(k-1)}}$, and we denote the set of patches at level $k$ as: $I_k^{rgb} = \{P_{h,w}^k \mid 1 \leq h \leq 2^{(k-1)}, \ 1 \leq w \leq 2^{(k-1)}\}$, where $P_{h,w}^k$ represents the patch located at the $h$-th row and $w$-th column of the partitioned image at scale $k$. We employ the CLIP model [34] to compute visual features for each patch. Specifically, for a given patch $P_{h,w}^k$, the corresponding visual feature is $v_{h,w}^k$. This process yields $k$ candidate features for each pixel, and then each patch $P_{h,w}^k \in I_k^{rgb}$ is independently processed using the Segment Anything Model (SAM) [35] to estimate the number of semantic instances it contains $n_{h,w}^k$, where $n_{h,w}^k$ denotes the number of instances detected within the patch $P_{h,w}^k$ at scale $k$.

In parallel, the depth image is divided into patches using the same scheme as the RGB image. For each depth patch $D_{h,w}^k$ that corresponds to the position of RGB patch $P_{h,w}^k$. We back-project the depth values into 3D space to generate a point cloud: $\mathcal{C}_{h,w}^k$. We then compute two geometric properties of each point cloud: the volume $V_{h,w}^k$ of the point cloud, density $\rho_{h,w}^k$ , defined as the number of points per unit volume: $\rho_{h,w}^k = \frac{|\mathcal{C}_{h,w}^k|}{V_{h,w}^k}$, where $|\mathcal{C}_{h,w}^k|$ denotes the total number of 3D points in the patch.

## A.2 Adaptive hierarchical feature selection

For each image pixel $p$ at hierarchical spatial level $l_s$, we select the CLIP feature from all candidate patches across scales that achieves the highest score under the corresponding scorer: $v_p^{l_s} = v_{h^*,w^*}^{k^*}, s_p^{l_s} = S_{h^*,w^*}^{l_s,k^*}$, and $(k^*, h^*, w^*) = \arg\max_{k,h,w:p \in P_{h,w}^k} S_{h,w}^{l_s,k}$. Where, $v_p^{l_s}$ denotes the image CLIP feature of pixel $p$ at level $l_s$, $s_{h,w}^{l_s,k}$ denotes the score assigned to patch $P_{h,w}^k$ at level $l_s$, and the constraint $p \in P_{h,w}^k$ ensures that only $p$ in patches are considered. After scoring, each pixel is back-projected into 3D and mapped to a voxel. For each semantic level, we keep only the feature with the highest confidence score in each voxel.

After scoring, each image pixel $p$ is back-projected into 3D space global position $x_p$ using the depth image and mapped to the corresponding voxel in the spatial map:

$$x_p = \text{BackProject}(p, I_t^{\text{depth}}(p), s_t) \in \mathbb{R}^3, \quad u_p = \left\lfloor \frac{x_p}{r} \right\rfloor \in \mathbb{Z}^3$$

where $r$ denotes the voxel resolution, and $\lfloor \cdot \rfloor$ indicates the element-wise floor operation used to discretize the 3D coordinate into voxel space. For each semantic level, if the voxel does not contain an existing feature, we directly store the current feature and its associated confidence score. If a feature already exists, we compare the new score with the stored score and retain the feature with the higher confidence. The voxel map is then updated according to the following rule:

$$\hat{v}_{u_p}^{l_s} = \begin{cases} v_p^{l_s}, & \text{if } u_p \notin \mathcal{M}_c \text{ or } s_p^{l_s} > \hat{s}_{u_p}^{l_s} \\ \hat{v}_{u_p}^{l_s}, & \text{otherwise} \end{cases}, \quad \hat{s}_{u_p}^{l_s} = \begin{cases} s_p^{l_s}, & \text{if } u_p \notin \mathcal{M}_c \text{ or } s_p^{l_s} > \hat{s}_{u_p}^{l_s} \\ \hat{s}_{u_p}^{l_s}, & \text{otherwise} \end{cases}$$

## A.3 Prompting

In the hierarchical landmarks generation, the complete prompt is as follows:

System: You are a helpful robot to find an object in an unknown environment.
User: Now that we need to find a/an {target}, please provide information about how it might be found in a private house. This information will be embedded with CLIP and must be useful for the robot to recognize the object.

- Provide which room it is likely to be found in.
- Provide where it is likely to be located within specific rooms. Please add the room type in front of the location.
- Provide what other items it is likely to be near.
- What is the probability of finding the {target} respectively around these landmarks? The sum of the probabilities of each level is equal to one.

At the same time, the output must meet the following requirements:

1. Output three related strings for each level. Each string should contain only one piece of information and avoid using "or" constructions.
2. Each piece of information should consist of only the most relevant phrases, not complete sentences. Keep the phrases simple and common; avoid uncertain words like "maybe".
3. Output two two-dimensional lists. For the landmarks string: The first dimension represents the information level, and the second dimension contains the three related strings for each level. For the Probability: The first dimension represents the information level, and the second dimension contains the three related probabilities for each level.
4. Output only the string and Probability, do not include any additional text.

In the generation of parameters $d_{\min}$ and $d_{\max}$ in the visibility map, the complete prompt is as follows:

System: You are a helpful robot to find an object in an unknown environment.
User: When we use `YOLOv7` and `GroundingDINO` to detect a/an {target} in a simulated mesh private house environment, the quality is poor. What is the best distance from the camera pose to the {target} to detect the {target}? The input images are $640 \times 480$ resolution.

1. Please output the distance in meters.
2. Please only output the distance range as a list: `[distance_min, distance_max]`, without any other text.

## A.4 Visibility map

As shown in Fig. 5, detection confidence depends on the pixel locations of the voxels in the image and the distance to the camera. Pixels near the image center and at moderate distances to the camera yield higher confidence, while peripheral or extreme-distance regions tend to have lower confidence due to reduced detectability. For each pixel $p$ in the RGBD image, we compute three components: the horizontal angle-based confidence $C_{\text{horizontal}}$, the vertical angle-based confidence $C_{\text{vertical}}$, and the distance-based confidence $C_d$. The detection confidence of pixel position $p$ is defined as:

$$C_{\text{horizontal}} = \cos^2\left(\frac{\theta_p}{\theta_{\text{hfov}}} \cdot \pi\right),$$

$$C_{\text{vertical}} = \cos^2\left(\frac{\phi_p}{\phi_{\text{vfov}}} \cdot \pi\right),$$

$$C_d = \begin{cases} 1, & \text{if } d_{\min} \le d_p \le d_{\max} \\ \exp\left(-\alpha \cdot \min\left((d_p - d_{\min})^2, \, (d_p - d_{\max})^2\right)\right), & \text{otherwise} \end{cases}$$

$$C_p = C_d \cdot C_{\text{horizontal}} \cdot C_{\text{vertical}}$$

Here, $\theta_p$ and $\phi_p$ denote the horizontal and vertical angles of pixel $p$, and $\theta_{\text{hfov}}$, $\phi_{\text{vfov}}$ are the horizontal and vertical FOV angles in radians. $C_d$ is computed from the pixel depth $d_p$ based on the optimal

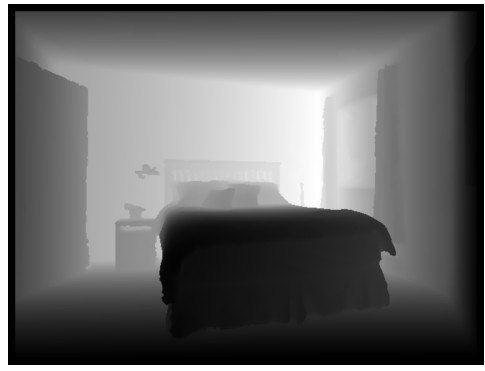
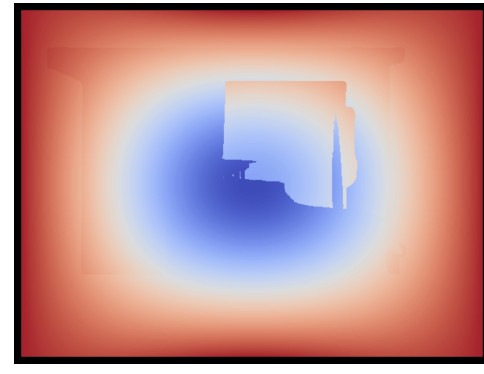

| (a) Input depth image | (b) Confidence value in FOV |
| --- | --- |

Figure 5: (a) is the input depth image. (b) shows the confidence computed from the depth image. Bluer regions indicate higher confidence, meaning the likelihood of an object being present is low if not detected there. Redder regions indicate lower confidence, implying that even if no object is detected in those areas, the probability of object presence remains relatively high.

range $[d_{\min}, d_{\max}]$ given by LLM A.3. The final confidence score $C_p$ means the confidence to detect the object at pixel position $p$.

Each pixel $p$ is back-projected into 3D space to obtain the corresponding voxel $u_p \in \mathbb{Z}^3$. The visibility map is then updated as:

$$\hat{p}^v_{u_p} = \begin{cases} 1 - C_p, & \text{if } u_p \notin \mathcal{P}^v \text{ or } 1 - C_p < \hat{p}^v_{u_p} \\ \hat{p}^v_{u_p}, & \text{otherwise} \end{cases}$$

## A.5 Path optimization

The optimization objective of the path planning problem is to find a frontier visiting order $\pi = [f_{\pi_1}, f_{\pi_2}, \ldots, f_{\pi_n}]$ that minimizes the expected search distance:

$$\pi^* = \operatorname*{argmin}_{\pi \in S_n} \sum_{i=1}^{n} \left( \sum_{k=1}^{i} d_{A^*}(f_{\pi_{k-1}}, f_{\pi_k}) \right) P_{\text{obs}}(f_{\pi_i})$$

Therefore the cost function $W(\pi)$ to evaluate the quality of a path $\pi$ can be defined as:

$$W(\pi) = \sum_{i=1}^{n} \left( \sum_{k=1}^{i} d_{A^*}(f_{\pi_{k-1}}, f_{\pi_k}) \right) P_{\text{obs}}(f_{\pi_i})$$

The simulated annealing algorithm is a probabilistic optimization algorithm that can be used to find an approximate solution to the path planning problem. The algorithm is inspired by the annealing process in metallurgy, where a material is heated and then cooled to remove defects and improve its properties. The algorithm works by iteratively exploring the solution space and accepting or rejecting new solutions based on their cost and a temperature parameter. Our implementation is based on the following steps:

1. **Initialization**: Set the initial and terminal temperature $T_0$ and $T_f$, the cooling rate $\alpha$, and the number of samples to simulate $N$. When the current temperature $T$ is greater than the terminal temperature $T_f$, the algorithm will continue to run.

2. **Iterative Process**: While the termination criterion is not met, for each sample, the algorithm will perform the following steps:

    (a) **Generating Neighbor Solution**: Generate a neighbor solution $\pi'$ from the current solution $\pi$ by applying three kinds of operations: swap, shift, or reverse.
    (1) swap: swap two points in the path. (2) shift: move a segment of the path to a different position. (3) reverse: reverse a segment of the path.

The repetition times of the operations are controlled by the temperature $T$, which decreases with time. Due to different operations having different degrees of impact on $\pi$, the probability of selecting each operation is different. And because the first point in the path is the starting point, it does not participate in this transformation.

(b) **Evaluation and Acceptance**: Evaluate the cost of the new path $W(\pi')$ and apply the Metropolis Criterion: Compare it with the cost of the current path $W(\pi)$. If $W(\pi') < W(\pi)$, accept the new path. Otherwise, accept it with a probability of $\exp\left(\frac{W(\pi) - W(\pi')}{T}\right)$.

(c) **Cooling**: Update the temperature $T$ by multiplying it with the cooling rate $\alpha$.

3. **Termination**: The algorithm ends when the temperature $T$ is less than the terminal temperature $T_f$. The final output is the sample with the lowest cost.

The algorithm is implemented in Python and uses the CuPy [52] library to accelerate the process. All the parameters are tuned to achieve a balance between the quality of the solution and the time taken to find it, for the path planning problem. On a laptop with Intel Core i5-12500H CPU, 16GB RAM, and NVIDIA GeForce RTX 2050 Laptop GPU, for a task of 10 frontiers, the algorithm takes about 0.2 seconds to solve the problem. A visualized example problem and the solution are shown in Fig. 7.

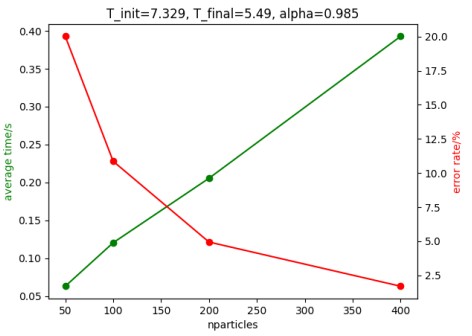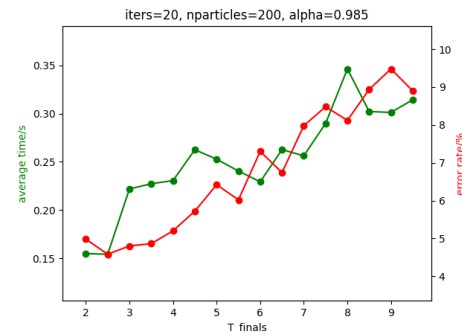

Figure 6: Error analysis of the algorithm for parameter tuning. All the results are based on 10 frontiers, over 50 different scenes, and each scene for 100 times. We take the solution obtained when the algorithm converges as the optimal solution. A solution is considered as an error if the cost is greater than 110% of which of the optimal solution. Between the performance of the algorithm and the time taken to find the solution, we take a balanced approach.

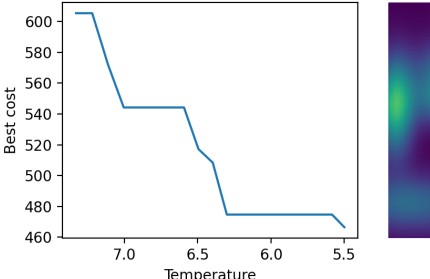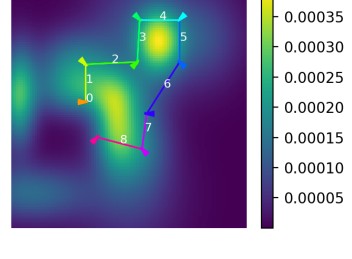

Figure 7: Result demo of the algorithm. The chart on the left shows the relationship between the cost and the temperature under a run. The chart on the right shows the path generated by the algorithm. The number on the lines indicates the order of visiting the frontiers. The arrow at each frontier is the orientation. The color of the background represents the distribution of the occurrence probability. The observation probability of a frontier is the sum of the weights of all points within a sector in front of the frontier, representing points within the FOV of the robot.

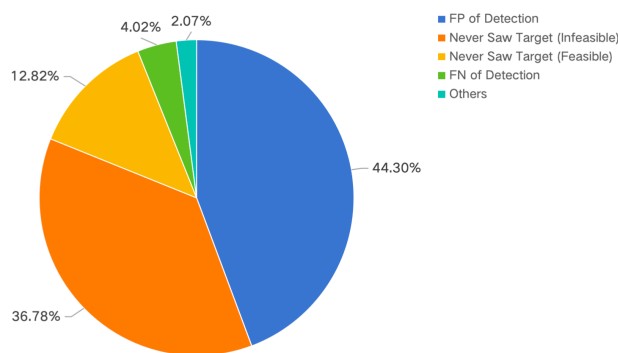

Figure 8: The proportion of different causes of failure in the HM3D dataset.

## A.6 Baselines

We evaluate our approach in comparison with a range of ZSON baselines, including several SOTA methods. ZSON [24] incorporates object category cues to enable object-aware navigation. CoWs [13] leverages CLIP features to extract semantic object information and directs exploration toward nearby frontiers. ESC [3] combines a semantic scene representation with commonsense reasoning to guide object search. L3MVN [14] utilizes large language models to infer exploration goals from a semantic map constructed by a pretrained detector. VLFM [7] adopts BLIP-2 [50] for vision-language alignment, using target object descriptions to prioritize exploration frontiers. VoroNav [28] introduces a navigation method based on Voronoi partitioning. InstructNav [9] supports agent navigation by converting the output instructions from the VLM into various value maps. However, this method directly relies on the VLM to reason in spatial contexts, which can lead to hallucinations and reduced navigation efficiency. GAMap [8] guides navigation by leveraging object parts and affordance attributes through an image-based, multi-scale scoring approach that effectively captures both geometric components and functional affordances. SG-Nav [12] guides navigation by leveraging an online 3D scene graph and LLM-based hierarchical chain-of-thought reasoning, enabling explainable, robust zero-shot object navigation with a graph-based re-perception mechanism to correct false positives. UniGoal [48] guides universal zero-shot navigation by leveraging graph-based scene–goal matching and LLM-driven multi-stage exploration, effectively unifying object, image, and text goals with coordinate projection, graph correction, and blacklist-aware reasoning.

## A.7 Error Analysis of HM3D

As shown in Fig. 8, the causes of failure cases can be primarily attributed to two factors. First, the current performance limitations of the open vocabulary detector, particularly due to false positives (FP) and false negatives (FN), account for a significant proportion, with these detection errors contributing to 48.32% of the failure cases. These errors result in the system either missing or incorrectly identifying the target object, which ultimately leads to failure in object navigation.

The second major factor stems from the existence of target objects located on a different floor than the agent's starting point in the HM3D dataset, which accounts for 36.78% of the failure cases. Although our 3D voxel map naturally supports modeling of spaces with varying heights, the limitations in stair recognition and local planner performance prevent successful traversal between floors, resulting in the inability to reach the target object on another floor.

Finally, only 12.82% of the failure cases are due to the target object being on the same floor but not found, which demonstrates the effectiveness and efficiency of our exploration module. The "other" category primarily involves cases where the target object was detected but the local planner was unable to navigate towards it, or where the number of steps exceeded the predefined limit.

## A.8 Additional ablation study

**Effectiveness of different LLMs** As shown in Table 6, the choice of LLM exerts only a marginal effect on BeliefMapNav: SR varies by only 1.0% (61.5–62.5) and SPL by 0.8 (30.9–31.7) across four architectures ranging from 8B to trillion-parameter scales. This stability arises because the framework

| Table 6: Impact of Different LLMs | | |
|---|---|---|
| LLMs | SR↑ | SPL↑ |
| Qwen3-32B [53] | 62.0 | **31.7** |
| gemini-2.5-flash [54] | 61.8 | 30.9 |
| Llama-3.3-8B-Instruct [55] | 61.5 | 31.4 |
| GPT-4o [36] | **62.5** | 31.6 |

| Table 7: Impact of the different image scales k. | | |
|---|---|---|
| Scale(k) | SR↑ | SPL↑ |
| 1 | 57.8 | 29.1 |
| 2 | 61.3 | 29.3 |
| 3 | 62.5 | **31.6** |
| 4 | **62.8** | 29.8 |

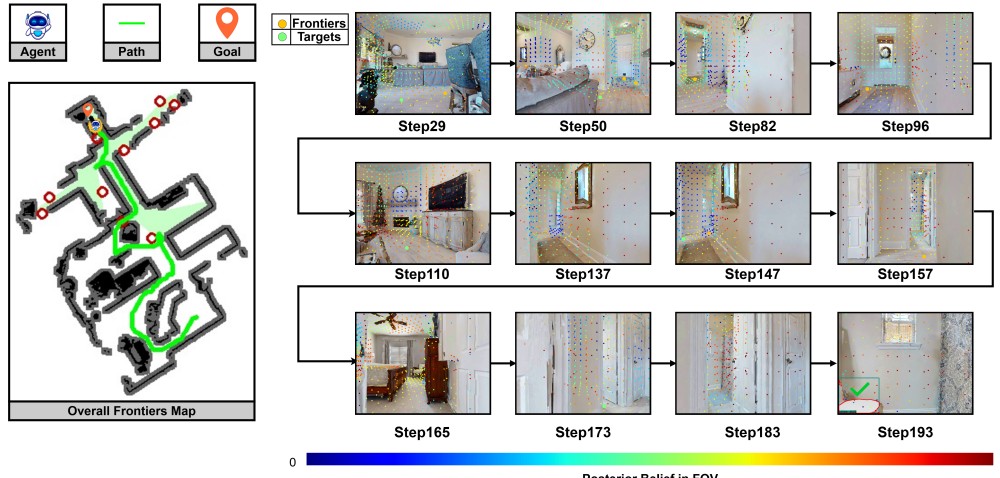

Figure 9: Visualization of the search process. The color of each point in the image represents the belief of object presence: redder points indicate higher belief, while bluer points indicate lower belief.

delegates metric spatial reasoning to the belief map, while the LLM is restricted to inferring landmark plausibilities from textual descriptions—an inference task that lies well within the capability ceiling of contemporary language models. Consequently, the pipeline exhibits negligible sensitivity to LLM specification, alleviating deployment constraints.

**Effectiveness of image scale** As shown in Table 7, increasing the Image Scales from 1 to 3 consistently improves both success metrics, with SR rising from 57.8 to a peak of 62.5 and SPL reaching its maximum value of 31.6. However, increasing the scale further to 4 yields only minimal gain in SR (from 62.5 to 62.8) but causes a notable reduction in SPL (from 31.6 to 29.8). This degradation in path efficiency is likely attributable to the introduction of noisy, fine-grained details at the small image scale, which negatively impacts target-oriented exploration.

## A.9 Visualization

As shown in Fig. 9, 10, and 11, we provide qualitative visualizations to highlight the behavior and strengths of BeliefMapNav. Fig. 12 shows the complete 3D voxel-based belief map, visibility map, and posterior belief map.

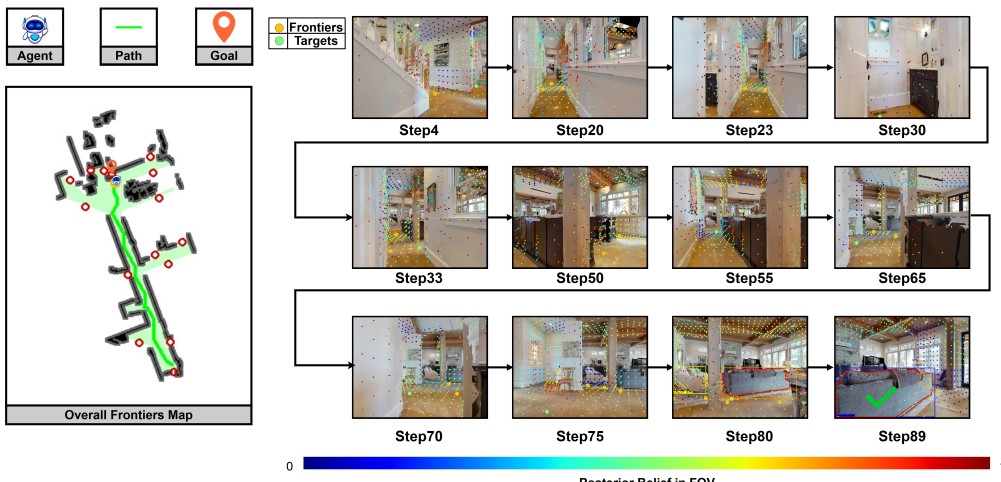

Figure 10: Visualization of the search process. The color of each point in the image represents the belief of object presence: redder points indicate higher belief, while bluer points indicate lower belief.

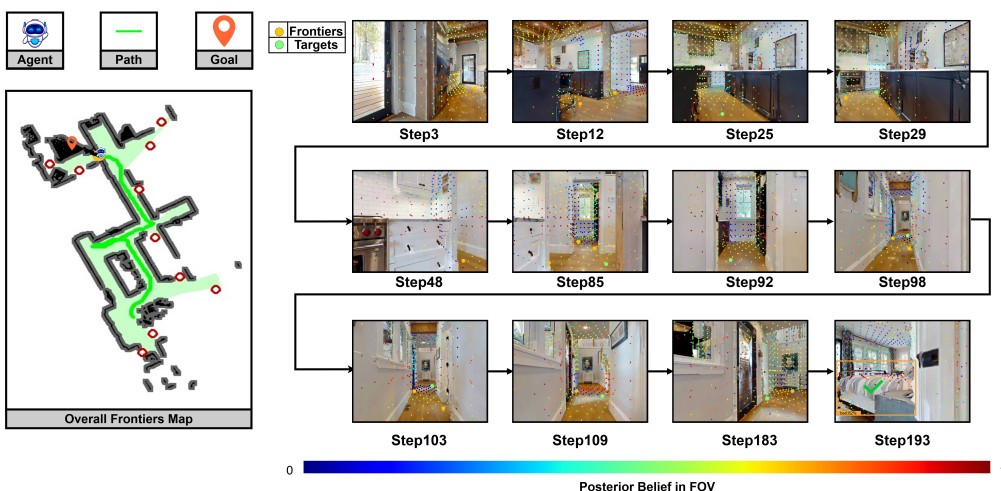

Figure 11: Visualization of the search process. The color of each point in the image represents the belief of object presence: redder points indicate higher belief, while bluer points indicate lower belief.

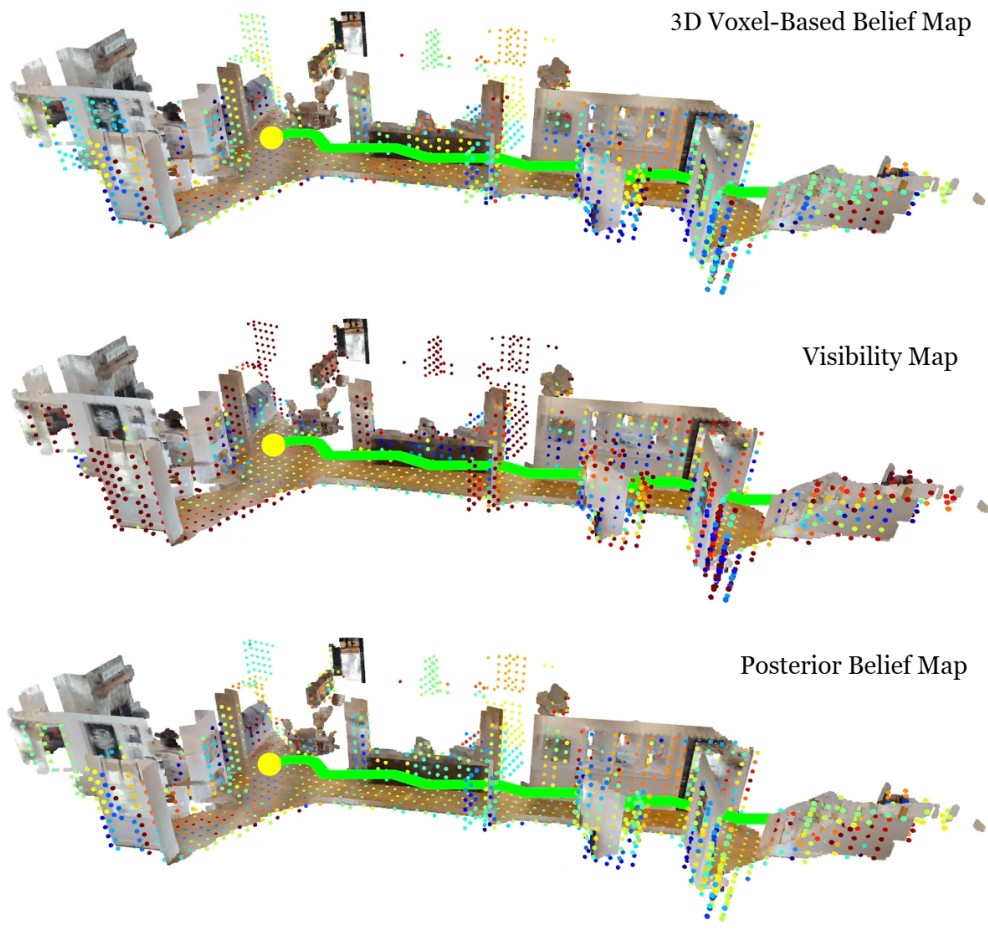

Figure 12: Visualization of the prior belief map, visibility map, and the posterior belief map, with an enlarged section highlighting the target object.

