# OpenReview forum: "BeliefMapNav: 3D Voxel-Based Belief Map for Zero-Shot Object Navigation"
_NeurIPS.cc/2025/Conference — NeurIPS 2025 poster_

### Official Review · Reviewer_yGEQ · 2025-06-30

**Clarity:** 2
**Significance:** 3
**Originality:** 2
**Rating:** 4
**Confidence:** 3

**Summary:**

This paper introduces BeliefMapNav, a novel zero-shot object navigation system that leverages a 3D voxel-based belief map to integrate hierarchical spatial semantics, commonsense reasoning from LLMs, and real-time observations. The proposed method aims to address limitations in existing approaches, such as greedy navigation strategies and poor spatial reasoning in LLMs/VLMs, by enabling fine-grained target localization and efficient path planning. Experiments on HM3D, MP3D, and HSSD benchmarks demonstrate state-of-the-art performance, with significant improvements in SR and SPL

**Questions:**

Please see weaknesses. I hope the authors can provide more explannations about this paper to address my concerns.

**Ethical Concerns:**

["NO or VERY MINOR ethics concerns only"]

**Final Justification:**

The authors' response has addressed my concerns. I have raised my score accordingly.

**Limitations:**

Yes.

**Paper Formatting Concerns:**

No.

**Quality:**

3

**Strengths And Weaknesses:**

Strengths
1. An interesting idea. This paper introduces 3D voxel-based map into emboddied navigation, which can leverage the observed spatial information to performance scene understanding and map-based planning.
2. Strong Results. The method achieves impressive improvements over some baselines, particularly on the SPL metric. This validates its effectiveness on path planning.
3. The paper is well-written with detailed figures to demonstrate some details of this proposed complex framework.

Weaknesses
1. I am not very familiar with 3D voxel technology, but as far as I know, voxels are suitable for efficiently processing real-world data and simplifying the representation of spatial information, but cannot directly and significantly improve system performance. Since this paper does not conduct experiments in real-world scenarios, is it really necessary to use voxels? In fact, most previous works have achieved good performance simply by constructing semantic maps. Furthermore, is the performance improvement in this paper actually due to the use of voxels, or is it because improving other modules.
2. I still don't understand why the proposed system achieves such an obvious improvement on SPL. The paper mentions that this is due to applying a distance cost-aware planner. However, previous methods like InstructNav also took distance into account to encourage efficient exploration. What do you think is the core reason for the improvement in SPL?
3. This paper still uses CLIP to help understand and locate object information. However, some of the latest works [1, 2] directly leverage VLMs (such as GPT-4o and Gemini) to process this information. Why did the authors not consider using these more powerful models to improve performance and generalization ability?

[1] InstructNav: Zero-shot System for Generic Instruction Navigation in Unexplored Environment.

[2] Affordances-Oriented Planning using Foundation Models for Continuous Vision-Language Navigation.

---

> ### Author Rebuttal · Authors · 2025-07-28
>
> > **[W1]:** is it really necessary to use voxels? is the performance improvement in this paper actually due to the use of voxels, or is it because improving other modules.
>
> **Answer:**
>
> **The necessity of using 3D voxel maps**
> * Object navigation is inherently a **3D spatial reasoning task.** Most prior methods based on 2D semantic maps suffer from intrinsic limitations—many target objects appear on different floors or at significantly varying heights. In such cases, 2D BEV semantic maps lose critical Z-axis information, leading to navigation confusion, misidentification, or even complete failure.
> * What we truly need is **accurate 3D spatial belief information.** For example, only 3D spatial information can accurately capture the occlusion relationships within the environment. This forms the foundation for applying ray casting to precisely estimate the belief of observing the target object from each camera pose. **The 3D voxel map just serves as a memory-efficient representation of such 3D spatial semantics.** The real performance gains in our system stem not from voxels themselves, but from the effective preservation of 3D spatial semantic information, which enables precise belief-based target localization and distance-optimized path planning across the full 3D environment.
>
> **Key contributors to performance gains**
>
> Our pipeline is built upon VLFM. We directly adopt its point navigation module, object detector, frontier selection strategy, and other supporting components without modification. The only additions we make are the **belief map** and the **belief-based path planning module.**
> Even with these two components alone, **our full method outperforms VLFM by +20.2% in SR and +8.6% in SPL**.
> Furthermore, when **using only the 3d voxel belief map (without path planning)**, the system **still achieves +7.7% SR and +0.69% SPL** improvement over the original VLFM baseline. Thus, representing semantic and target object belief information within a 3D voxel map preserves the full spatial structure of the environment, providing a reliable foundation for downstream 3D tasks like navigation.
>
> * **table 1: blation study of 3d voxel- based belief map and path planning**
>
> | Method| SR↑| SPL↑|
> |-|-|-|
> | VLFM[1]| 52.0| 29.1|
> | VLFM+belief map| 56.0| 29.3|
> | BeliefMapNav(VLFM + belief map + path planning) | **62.5** | **31.6** |
>
> > **[W2]:** why the proposed system achieves such an obvious improvement on SPL. The paper mentions that this is due to applying a distance cost-aware planner. However, previous methods like InstructNav also took distance into account to encourage efficient exploration. What do you think is the core reason for the improvement in SPL?
> >
> **Answer:**
>
> The significant SPL improvement of BeliefMapNav is primarily driven by its **1)distance-aware planner** and **2)the integration of prior belief with real-time observation feedback**. Although InstructNav[3] also takes distance into account to reduce redundant exploration, it does not perform optimization for the exploration distance. In contrast, BeliefMapNav **continuously refines its search strategy** by performing real-time path optimization based on the dynamically updated belief map throughout the distance-aware planner.The reasons for BeliefMapNav’s superior performance in SPL are as follows:
> * **Dynamic Search Efficiency**
> 	*	**InstructNav** tends to prioritize immediate gains **using a greedy strategy and treats distance merely as a static factor in its value map**, without performing explicit optimization for search efficiency. As a result, it often suffers from inefficient back-and-forth movements and cannot guarantee effective exploration, especially in complex environments.
> 	*	**BeliefMapNav**, on the other hand, **employs a distance-aware planner to globally optimize the search path**, reducing redundant revisits and significantly improving search efficiency.
> 	*	In table 2, without a distance-aware planner, **the SR drops 10.4%,SPL drops 7.9%.**
> *	**Effective Uncertainty Management**
> 	*	**InstructNav** emphasizes short-term rewards and **lacks mechanisms for managing exploration uncertainty**, often leading to inefficient and suboptimal path choices.
> 	*	**BeliefMapNav** dynamically evaluates exploration regions by **combining the posterior belief map with a visibility map**, allowing it to avoid exploring well-observed areas and focus on high-value regions.
> 	*	In Table 2, without combining the posterior belief map with a visibility map, **the SR decreases 8.48% and SPL drops 11.4%.**
>
> * **Table 2: Ablation study of Visibility Map and distance-aware planner**
>
> | Method                          | SR↑  | SPL↑ |
> |---------------------------------|------|------|
> | BeliefMapNav w/o Planner        | 56.0 | 29.1 |
> | BeliefMapNav w/o Visibility Map | 57.2 | 28.0 |
> | BeliefMapNav                    | 62.5 | 31.6 |
>
> > **[W3]:** Why did the authors not consider using these more powerful models to improve performance and generalization ability?
> >
> **Answer:**
>
> **Our task is fundamentally different from that in [2]—their task is more suitable for VLMs, whereas ours is not.**
> * The task in [2] is Vision-and-Language Navigation (VLN), where the agent follows detailed instructions to reach a target location. Since these instructions provide dense and explicit guidance, **the task mainly requires understanding the current scene and matching it to a specific step in the instruction, which VLMs handle well.**
>
> * In contrast, object navigation involves sparse prompts and large spatial uncertainty. It **requires building a global memory of the environment and performing complex 3D reasoning to locate the object, which are areas where VLMs perform poorly.** Therefore, we do not rely on VLMs for spatial inference in our method.
>
> **VLMs in Object navigation**
>
> * Although VLMs demonstrate strong capabilities in visual reasoning, they exhibit two major limitations in object navigation tasks.
>
>     1. **Lack of global spatial understanding**: Without explicit spatial modeling, VLMs struggle to reason about the likely location of the target object across the entire environment. As exploration progresses, they fail to maintain a globally consistent belief about where the target might be.
>
>     2. **Limited spatial memory method**: VLMs typically process single-frame or short-context inputs and lack mechanisms for building and maintaining persistent spatial memory. This makes it difficult to integrate multi-view observations and reconstruct a coherent spatial layout during exploration.
>
> **VLMs in InstructNav:**
> 1. **Limited Spatial Memory:** InstructNav leverages a Trajectory Value Map to avoid revisiting explored areas, but it lacks true object-level spatial memory.
> 2. **No Global Spatial Understanding:** InstructNav does not build or maintain a global map (topological or geometric) of the environment. All reasoning is local, relying on momentary observations and commonsense rules from LLMs.
> 3. **VLM Predictions Are Local and Stateless**: While VLM assists with directional prediction via the Intuition Value Map, its inference is based only on the current panoramic input.
>
> **BeliefMapNav's advantages in Object navigation:**
>
> BeliefMapNav is specifically designed to address the key limitations of VLMs in object navigation, effectively overcoming challenges such as spatial memory, global understanding, and 3D reasoning:
>
> 1. **Global spatial understanding:** belief map combines prior knowledge with real-time observations, dynamically inferring the posterior distribution of the target throughout the space, thereby possessing global consistency reasoning capabilities and significantly improving the accuracy of target localization and path planning.
> 2. **3d voxel-based belief map for spatial memory:** belief map represents the spatial structure and semantic information of the entire environment through 3D voxel, enabling the construction and update of spatial memory;
> 3. **Target reasoning with hierarchical semantic context:** BeliefMapNav avoids the spatial reasoning limitations of VLMs by utilizing LLMs to perform semantic-level landmark inference, a task they handle with much higher reliability and accuracy. This approach ensures more consistent target localization while leveraging LLMs in the domain where they perform best—structured, hierarchical semantic reasoning.
>
> **Experiment evidence:**
> * **Performance comparation:** as shwon in table 3, **InstructNav performs notably worse than ours on object navigation,** especially in SPL. Its SR gains come from meticulous searching and better detection, but it lacks efficient exploration.
> * **Spatial reasoning:** As shown in Table 4, using various strong LLMs has little impact on overall performance, indicating that our method effectively leverages their commonsense and task reasoning while avoiding VLMs’ spatial limitations. **The consistently strong results across LLMs also demonstrate the robustness of our approach.**
>
> **Table 3: performance of InstructNav and BeliefmapNav on HM3D**
> | Method       | SR↑(HM3D) | SPL↑(HM3D) |
> | ------------ | -------- | --------- |
> | InstructNav[3]  | 58.0     | 20.9      |
> | BeliefmapNav | **61.4** | **30.6**  |
>
> **Table 4: performance of Different LLMs on 400 Random Episodes in HM3D**
> |         LLMs          |    SR↑    |   SPL↑    |
> |:---------------------:|:--------:|:--------:|
> |       Qwen3-32B       |   62.0   | **31.7** |
> |   gemini-2.5-flash    |   61.8   |   30.9   |
> | Llama-3.3-8B-Instruct |   61.5   |   31.4   |
> | kimi-k2-0711-preview  |   62.3   |   31.2   |
> |        GPT-4o         | **62.5** |   31.6   |
>
> [1] Yokoyama, Naoki, et al. "Vlfm: Vision-language frontier maps for zero-shot semantic navigation."
>
> [2] Chen, Jiaqi, et al. "Affordances-oriented planning using foundation models for continuous vision-language navigation."
>
> [3] Long, Yuxing, et al. "Instructnav: Zero-shot system for generic instruction navigation in unexplored environment."

---

> > ### Comment · Reviewer_yGEQ · 2025-08-04
> >
> > Thank you for your response, which has addressed my concerns. I have raised my score accordingly.

---

### Official Review · Reviewer_RDXG · 2025-06-30

**Clarity:** 2
**Significance:** 2
**Originality:** 3
**Rating:** 4
**Confidence:** 5

**Summary:**

This paper proposes BeliefMapNav, which builds a 3D voxel-based representation and a belief-based planning module to achieve efficient navigation. The authors propose to integrate hierarchical spatial-visual features with LLM-derived commonsense and design a novel planner for the system. Experimental results on several benchmarks demonstrate the effectiveness of the proposed method.

**Questions:**

1. As a 2D problem, what is the main advantage of a 3D voxel map?
2. How does different LLM affects the performance of this work?
3. The most interesting point of this paper is that the proposed representation is not semantic-only, but distance-aware. So it may benefit path planning to achieve better navigation efficiency in terms of SPL. The authors should clarify what is the main difference between belief map-based representation and previous semantic map or scene graph-based representation.

**Ethical Concerns:**

["NO or VERY MINOR ethics concerns only"]

**Final Justification:**

The authors' rebuttal solve some of my concerns. However, I still have two questions / suggestions:
1. For a fair comparison, this paper should compare with all zero-shot navigation methods, regardless of their representations. So there is no reason to exclude methods like SG-Nav and UniGoal. A more comprehensive comparison is required in revised version.
2. I understand the authors' claim that navigation is a 3D problem (Height-Sensitiv, Cross-floor, ...). However, in current benchmarks (only one floor, and the task does not require fine-grained path planning to avoid collision), all these aspects cannot be fully demonstrated by experiments. If the authors do not provide experiments to validate the 3D map's capability on these more challenging settings, at least they should make an ablation study to compare 3D map with 2D map, to validate the 3D representation is effective even in current relatively simple environments.

**Limitations:**

yes

**Quality:**

2

**Strengths And Weaknesses:**

# Strength:
The belief map is integrated into the path planning system. It also achieves good performance in terms of SPL, which validates its effectiveness.


# Weakness:
1. Lack of comparison with state-of-the-art methods including SG-Nav (NeurIPS 2024) and UniGoal (CVPR 2025) in Table 1. On MP3D, the SR of the proposed method is lower than SOTA.
2. The proposed belief map-based planning method benefits path planning. But in terms of design, it still leverages a similar GlobalPolicy-->LocalPolicy pipeline to generate the actions. The authors should clarify why the proposed method can improve the navigation efficiency.
3. Comparison with other path planning method, like frontier-based exploration, is expected.

---

> ### Author Rebuttal · Authors · 2025-07-28
>
> > **[W1]:** Lack of comparison with state-of-the-art methods including SG-Nav (NeurIPS 2024) and UniGoal (CVPR 2025) in Table 1. On MP3D, the SR of the proposed method is lower than SOTA.
>
> **Answer:**
>
> We identified these two excellent methods early on, but as both rely on scene graph representations, we excluded them from direct comparison. If there is a chance, **I will definitely include them in the experimental table.**  Nonetheless, As shown in table 2 our **approach still demonstrates clear performance gains over them**.
>
> * **Experimental data analysis:**
> As shown in the table, BeliefMapNav significantly outperforms prior **SOTA methods in both SR and SPL on HM3D**, highlighting its strong capability in complex and high-quality environments. While its SR on MP3D is slightly lower, it **still achieves a notably higher SPL**, demonstrating more efficient and purposeful navigation compared to other methods.
>     *	On HM3D, BeliefMapNav achieves 61.4% SR and 30.6% SPL, **significantly surpassing SG-Nav and UniGoal** by a large margin in both success rate and path efficiency(SPL). This demonstrates the strength of our approach in accurately reasoning over spatial structure and planning globally optimal paths.
>     *	On MP3D, while BeliefMapNav’s SR is slightly lower, it still yields the highest SPL (17.6) among all methods, indicating that it navigates with the most efficient and purposeful trajectories, even under more noisy or degraded conditions.
>
> * **Limited Improvement on MP3D**
> The slightly lower SR of BeliefMapNav on MP3D can be **attributed to two dataset-specific limitations, rather than a weakness of the method itself:**
>
>     * **Low mesh quality hinders object recognition:** MP3D’s poor mesh makes targets hard to detect. SG-Nav mitigates this with **re-perception**, improving SR. BeliefMapNav instead localizes targets via belief map–guided planning, and despite not enhancing detection, still leads in SPL through optimized exploration.
>
>     * **Broken surfaces and holes distort observation beliefs**: MP3D’s mesh flaws disrupt BeliefMapNav’s ray tracing, causing false high beliefs in non-traversable but visually reachable areas. This leads to some inefficiency and slight SPL drop, yet BeliefMapNav still achieves strong SPL gains, demonstrating robustness to noise.
> **Table 1: comparisons with SG-Nav and UniGoal**
> | Method| SR↑(HM3D) | SPL↑(HM3D) | SR↑(MP3D) | SPL↑(MP3D) |
> |-|-|-|-|-|
> |SG-Nav[2]|54.0| 24.9| 40.2 |16.0|
> |UniGoal[4]| 54.5| 25.1| **41.0** |16.4|
> |BeliefmapNav | **61.4**| **30.6**|37.3|**17.6**|
> > **[W2]:** The proposed belief map-based planning method benefits path planning. But in terms of design, it still leverages a similar GlobalPolicy-->LocalPolicy pipeline to generate the actions. The authors should clarify why the proposed method can improve the navigation efficiency.
>
> **Answer:**
> BeliefMapNav dramatically outperforms traditional greedy GlobalPolicy–>LocalPolicy pipelines, and the advantages are clear:
> * **In Global Optimization**: Unlike the basic “best next frontier” approach used by others, BeliefMapNav employs GPU-accelerated simulated annealing to optimize the entire path, minimizing the average exploration cost based on the posteriori belief map. This results in a far more efficient exploration.
> * **Real-Time Adaptive Planning**: The global path is recalculated at every frame to adapt to the dynamically evolving posteriori belief map, which reflects the latest observations. In the early stage, this enables the planner to make exploratory decisions under uncertainty. As the belief map converges, the planned path gradually approaches the globally optimal trajectory, balancing exploration and exploitation while avoiding redundant movement.
> * **Proven Performance Gains:** Our experiments show that removing the global belief-based planning results in a sharp performance drop (**SR down by 10.4%, SPL down by 7.27%**). This proves that our global optimization is essential for improving navigation efficiency and avoiding the pitfalls of short-sighted local planning.
>
> **In conclusion, BeliefMapNav’s global optimization and real-time recalculations ensure superior navigation efficiency and robust exploration—advantages traditional frontier-based methods cannot match.**
>
> > **[W3]:** Comparison with other path planning method, like frontier-based exploration, is expected.
>
> **Answer:**
> In our ablations (Table 2 & 5), **we have already conducted the comparison.**  For **Random frontier selection (SR: 21.5, SPL: 10.8)** or choosing the **highest-probability frontier (SR: 56.0, SPL: 29.3)** performs significantly worse (**our method with SR: 61.5, SPL: 31.6**), highlighting the advantage of our global optimization, which considers full paths and avoids frontier-based inefficiencies.
>
> > **[Q1]:** As a 2D problem, what is the main advantage of a 3D voxel map?
>
> **Answer:**
> **This is not a 2D problem—it is inherently a 3D task.** Real-world environments are three-dimensional, and tasks like locating objects across floors or at varying heights cannot be fully addressed with 2D maps. A 3D voxel map naturally handles these challenges by preserving spatial structure and enabling accurate reasoning across all dimensions.
>
> * **Handling Height-Sensitive Environments**: Targets may appear at vastly different heights (e.g., 0.3 m vs. 3.0 m). 2D projections lose height semantics, causing misidentification, while 3D voxel maps retain centimeter-level resolution for accurate localization.
>
> * **Cross-Floor and Height Variation**: A 2D map cannot distinguish between identical areas at different heights—e.g., “the desk in the bedroom on the second floor” vs. “the desk on the first floor.” In contrast, 3D voxel maps explicitly encode the Z-axis, enabling accurate cross-floor search, which is not feasible with flat 2D representations.
>
> * **robust cross-embodiment performance across different robot heights:** 3D voxel maps preserve height information, enabling compatibility with robots of varying heights and traversability constraints.
>
> * **Occlusion and Visibility**: 3D voxel maps enable ray tracing to accurately model occlusion, making it possible to **determine whether a given viewpoint can truly observe a potential object**, it is an essential factor in realistic visibility estimation. In contrast, 2D maps lack this capability and often make incorrect assumptions about what is observable.
> * **Future Expansion**: The 3D voxel map is not limited to navigation; it can be seamlessly adapted for tasks such as manipulation and multi-floor inspections. This cross-task versatility eliminates the need for re-mapping, which would be necessary with 2D maps for different applications.
> *  **conclusion:** The 3D voxel map offers the precision, adaptability, and scalability needed for tasks involving height variations, complex spatial reasoning, and efficient target object localization. These capabilities simply cannot be achieved with 2D maps, making the 3D voxel approach indispensable for real-world applications in multi-floor environments.
>
> > **[Q2]:** How does different LLM affects the performance of this work?
>
> **Answer:**
>
> **Table 2: performance of Different LLMs on 400 Random Episodes in HM3D**
> | LLMs | SR |SPL|
> |:-:|:-:|:-:|
> |Qwen3-32B|62.0|**31.7** |
> |gemini-2.5-flash|61.8|30.9|
> | Llama-3.3-8B-Instruct |61.5|31.4|
> | kimi-k2-0711-preview  |62.3|31.2|
> | GPT-4o| **62.5** |31.6|
> Table 2 shows that different LLMs have minimal impact on BeliefMapNav’s performance. This is because we don’t rely on LLMs for spatial reasoning—instead, they infer landmarks and probabilities from text and commonsense, a task well within current LLM capabilities. This also **highlights the method’s robustness across LLMs choices.**
>
> > **[Q3]:** The authors should clarify what is the main difference between belief map-based representation and previous semantic map or scene graph-based representation.
>
> **Answer:**
> The table below compares our 3D voxel-based belief map with prior methods. **Our approach shows clear advantages across multiple dimensions, leading to SOTA performance on several benchmarks.**
> |Attribute| Previous Semantic Map \[1]| Scene Graph-Based \[2]|BEV Value-Based \[3]| 3D Voxel-Based Belief Map (Ours)| 3D Belief Map’s Advantage over This Attribute|
> |-|-|-|-|-|-|
> |**Representation**|Category-level occupancy grid or object-centric maps from detection|Topological representation (e.g. Nodes = objects/rooms, edges = relations)| Bird’s-eye view 2D grid encoding target value maps from pixel semantics| Dense 3D voxel grid storing continuous belief values of target presence | more accurate target object localization by Integrating dense 3D geometry, hierarchical semantics, and LLM commonsense|
> | **Semantic Granularity** | Coarse (category-level)|discrete: Graph nodes with edges| Mid-grained: misaligned spatial and image semantics at pixel level|Fine-grained: 3D spatial via voxel-level uncertainty| Finer, with 3D structure, observation likelihood dynamic|
> | **Spatial Reasoning**| Topological or 2D grid| Topological reasoning over discrete relations| Image-based reasoning| Multi-level (scene → region → object) + LLM commonsense + dynamic observation likelihood| Full 3D dynamic volumetric reasoning, capturing height & depth|
> |**Key Limitations**|Semantic loss from detection failures; misaligned spatial scales|Graph sparsity misses fine-grained cues; fragile to detection errors| Ambiguous positional values; lacks height & 3D context;misaligned spatial scales |High computation| Overcomes semantic loss, sparsity, and 2D limitations, and cover the High computation with Embedding codebook |
>
> [1] "L3mvn: Leveraging large language models for visual target navigation."
>
> [2]"Sg-nav: Online 3d scene graph prompting for llm-based zero-shot object navigation."
>
> [3]"Vlfm: Vision-language frontier maps for zero-shot semantic navigation."
>
> [4] Yin, Hang, et al. "Unigoal: Towards universal zero-shot goal-oriented navigation."

---

> ### Author Response · Authors · 2025-08-05
> **Reviewer Discussion Invitation and Summary of Key Responses**
>
> Dear Area Chairs and Reviewers,
>
> We would first like to thank all the reviewers for their thoughtful feedback sincerely — your comments are extremely valuable to us. We have carefully addressed every concern raised through detailed clarifications and additional experiments.
>
> **We are encouraged to see that Reviewer yGEQ felt our clarifications resolved their concerns** and chose to raise their score. **As the reviewer–author discussion period approaches its deadline (August 6th)**, we warmly invite more reviewers to engage in the discussion, which would greatly help us further improve the quality of our work. If our responses have helped address your concerns, we would be truly grateful if you would consider updating your score accordingly.
>
> Below, we provide a summary of our responses to each reviewer’s specific concerns:
>
> **Reviewer kq75**:
> 1. We clarified the differences and advantages of our approach over VLFM from multiple perspectives, achieving significantly better performance under the same pipeline **(SR: +20.2%, SPL: +8.6%).**
> 2. Compared to SG-Nav and UniGoal, our method consistently outperforms on HM3D **(SR: +12.6%, SPL: +21.9%)**. On MP3D, while SR is slightly lower **(–9%)**, we still achieve a notable gain in SPL **(+7.3%)**. The detailed analysis is provided in the response; **the lower performance is likely due to dataset characteristics rather than limitations of our method.**
> 3. We provide evidence from multiple aspects showing that integrating only the top three high-confidence patches into the voxel map does **not** lead to unstable mapping or limited semantic generalization. Furthermore, experiments show that adding more patches does **not** yield meaningful performance gains.
>
> **Reviewer RDXG:**
> 1. Coincidentally, both SG-Nav and UniGoal were mentioned by different reviewers, and we have included comparisons with both in our response.
> 2. We clarify how the proposed method improves navigation efficiency from the perspectives of global optimization, real-time adaptive planning, and empirical results **(SR: +11.6%, SPL: +7.85%).**
> 3. Additionally, the paper **includes ablation studies using alternative planners.**
>
> **Reviewer cjFY:**
> 1. We provide theoretical support for our system design through **optimization modeling** and **problem decomposition** of the object navigation task.
> 2. We added a set of experiments to demonstrate the importance of the **planner** and **belief map**, as well as their contributions to performance improvement **(belief map: SR +7.7%, SPL +0.69%; planner + belief map: SR +20.2%, SPL +8.6%).**
>
> **Reviewer yGEQ:**
> 1. We analyze the limitations of directly using VLMs for object navigation from multiple perspectives, and highlight the advantages of incorporating a voxel map.
> 2. We also explain the core reason behind the observed improvement in SPL.
>
> Your feedback is highly appreciated and plays a crucial role in shaping our work. We remain fully open to addressing any remaining questions and welcome further discussion.
>
> Sincerely,
> Authors

---

> ### Author Response · Authors · 2025-08-06
>
> Dear Reviewer,
>
> We noticed that your rating is currently not visible, and we are not sure whether our response has sufficiently addressed your concerns. If there are any remaining issues or aspects that you feel require further clarification, we would greatly appreciate your feedback.
>
> Your feedback is extremely valuable to us, and we thank you again for your time and effort in reviewing our work.

---

> > ### Comment · Reviewer_RDXG · 2025-08-06
> >
> > Thanks for the authors' rebuttal. It solves many of my concerns. However, I still have two questions / suggestions:
> >
> > 1. For a fair comparison, this paper should compare with all zero-shot navigation methods, regardless of their representations. So there is no reason to exclude methods like SG-Nav and UniGoal. A more comprehensive comparison is required.
> >
> > 2. I understand the authors' claim that navigation is a 3D problem (Height-Sensitiv, Cross-floor, ...). However, in current benchmarks (only one floor, and the task does not require fine-grained path planning to avoid collision), all these aspects cannot be fully demonstrated by experiments. If the authors do not provide experiments to validate the 3D map's capability on these more challenging settings, at least they should make an ablation study to compare 3D map with 2D map, to validate the 3D representation is effective even in current relatively simple environments.
> >
> > It is expected the authors can solve these problems in a revised version of the paper.

---

> ### Author Response · Authors · 2025-08-06
>
> Thank you again for your thoughtful question and suggestion. We apologize for the oversight and for not addressing these two points more clearly in our initial response. However, we are able to provide clear and satisfactory answers to both of these questions.
>
> > **Q1 comparison of SG-Nav and Unigoal**
>
> **Answer:**
>
> First, we would like to highlight two key points regarding the comparison:
> 1. BeliefMapNav **outperforms the two methods in most metrics across both datasets.** Specifically, it achieves clearly superior SR and SPL on HM3D, and also shows a significant advantage in SPL on MP3D. The only metric where it slightly underperforms is SR on MP3D.
> 2. The relatively lower SR on MP3D is primarily due to differences in the detector module, **rather than the core contributions of our method—the belief map representation and its associated planner.** In the error analysis, we observe that **the detector’s false positive rate (FP) in MP3D failure cases is significantly higher (73.9%) compared to that in HM3D (44.3%).** This suggests that the **performance gap is not caused by our proposed planning framework,** but by the detection component, which could be improved independently.
>
> Secondly, we sincerely apologize for the oversight in not including these two important works in our comparison. **If the paper is fortunate enough to be accepted, we will make it a priority to incorporate both methods into our experimental evaluation.** That being said, we would like to respectfully emphasize that **our method demonstrates clear performance advantages compared to these works, as can be observed from the existing metrics.**
>
> Lastly, in early experiments, **SG-Nav achieved SR: 41.7 and SPL: 20.4 on HM3D,** which deviates from the results reported in the paper **(SR: 54.0, SPL: 24.9).** This discrepancy was also one of the reasons we did not include it in our comparisons.
>
> > **Q2: Advantages of 3D Map over 2D Map**
>
> **Answer:**
>
> **For the comparison of 2D map and 3D map:**
>
> We apologize for the oversight. **As addressed in our response to another reviewer, we have already compared the performance differences between the 3D map and the 2D map.**
>
> **table 1: blation study of 3d voxel- based belief map and path planning**
> | Method                                          | SR↑      | SPL↑     |
> | ----------------------------------------------- | -------- | -------- |
> | VLFM[1](2D map)                                        | 52.0     | 29.1     |
> | VLFM+belief map (3D map)                                 | 56.0     | 29.3     |
> | BeliefMapNav(VLFM + belief map + path planning) | **62.5** | **31.6** |
> **Our pipeline is built upon VLFM.** We directly adopt its point navigation module, object detector, frontier selection strategy, and other supporting components without modification. when **using only the 3d voxel belief map (without path planning)**, the system **still achieves +7.7% SR and +0.69% SPL** improvement over the original VLFM baseline. Thus, representing **semantic and target object belief information within a 3D voxel map preserves the full spatial structure of the environment**, providing a reliable foundation for downstream 3D tasks like navigation.
>
> **For the multi floor:**
> 1. In the HM3D dataset, **28.1% of the episodes involve scenarios where the agent’s starting position and the target object are located on different floors.** Similarly, in the MP3D dataset, **this occurs in 15.2% of the episodes.** These cross-floor navigation cases are also analyzed in detail in the error analysis section of the appendix.
> 2. Although our current low-level point navigation planner does not explicitly support active stair detection and navigation, but some downstairs scenarios in ablation studies, the 3D voxel-based belief map can still guide the agent to the target **without any modification or map switching.** Excluding these cases leads to a slight performance drop **(SR: 62.5 → 60.5, SPL: 31.6 → 31.5)**, **demonstrating the inherent compatibility of the belief map with multi-floor search, even without additional engineering.**
> 3. The lack of cross-floor navigation is due to the absence of a stair detection module—**an engineering limitation rather than a flaw of the belief map and its planner.** **Both components naturally support multi-floor navigation,** and we plan to add stair detection in future work to demonstrate the 3D belief map’s potential better. This limitation does not affect the core contributions of our method.
>
>
> **We look forward to your new feedback. If there are any remaining questions or aspects that require further clarification, we would greatly appreciate your feedback.**
>
> [1]"Vlfm: Vision-language frontier maps for zero-shot semantic navigation."

---

> > ### Comment · Reviewer_RDXG · 2025-08-07
> >
> > Thanks for the authors' timely reply. I will raise my score.

---

### Official Review · Reviewer_cjFY · 2025-07-02

**Clarity:** 3
**Significance:** 3
**Originality:** 3
**Rating:** 4
**Confidence:** 4

**Summary:**

In this paper, the authors propose a zero-shot object navigation system that constructs a 3D voxel-based belief map to estimate the target's likely location. This belief map integrates semantic priors from large language models (LLMs), visual features from CLIP, hierarchical spatial information, and real-time observations. Based on the belief map, the authors design an A*-based planning module to enhance exploration efficiency. Experimental results demonstrate that the proposed system achieves state-of-the-art performance.

**Questions:**

see Weaknesses

**Ethical Concerns:**

["NO or VERY MINOR ethics concerns only"]

**Final Justification:**

I acknowledge and appreciate the detailed design considerations presented by the authors. Nevertheless, considering the highly engineering-focused nature of the Object Goal Navigation task, I will maintain my original borderline accept evaluation.

**Limitations:**

yes

**Quality:**

3

**Strengths And Weaknesses:**

**Strengths**:
1. The paper is well written and easy to follow.
2. The design of each module in the paper is well-justified, and the authors provide detailed explanations for each component in the supplementary material. The method of fusing the visibility map and belief map into a posterior map is both reasonable and practically valuable. The use of an A*-based path planning strategy makes more efficient use of the computed belief compared to a greedy approach.
3. The experimental section is thorough, and the proposed system demonstrates strong performance. Moreover, the ablation studies effectively support the authors' claims and validate the effectiveness of each module.

**Weaknesses**：
1. It would be helpful to understand how much the planner contributes to the final results shown in Table 1. I suggest that the authors include an additional ablation study to isolate the impact of the planner and provide a clear explanation of the results.
2. A significant amount of detailed content is placed in the supplementary material rather than in the main text, which may be unfriendly to readers who are not familiar with this area. Including more key information in the main paper would improve clarity and accessibility.
3. Although the system achieves impressive results, some design choices in the paper, such as the posterior fusion strategy and the various scorers, appear somewhat heuristic or empirically driven, lacking sufficient theoretical justification.

---

> ### Author Rebuttal · Authors · 2025-07-29
>
> > **[W1]:** Please clarify the planner’s individual contribution in Table 1 by adding an ablation study to isolate its impact and explain the results clearly.
>
> **Answer:**
> Thank you for the suggestion. While our ablation already highlights the planner’s role, we understand the need to clarify the belief map’s independent contribution(In Table 2 of our paper). We initially did not separate the planner, as it is an essential, tightly coupled component of BeliefMapNav that directly serves the belief map.
>
> To address this, we added experiments on HM3D (Table 1) to isolate the contributions of the belief map and planner. Since our system is built on VLFM [1], with all other modules unchanged, it serves as a natural baseline—allowing us to clearly attribute performance gains to our two additions: the 3D voxel-based belief map and belief-guided planner
>
> **table 1: Ablation study of 3d voxel-based belief map and path planning on HM3D with 400 samples**
>
> |Method| SR↑| SPL↑|
> |-|-|-|
> | VLFM[1]| 52.0| 29.1|
> | VLFM+Belief Map| 56.0| 29.3|
> | BeliefMapNav(VLFM + Belief Map + Path Planning) | **62.5** | **31.6** |
>
> With just the belief map and planner, our method improves SR by +20.2% and SPL by +8.6% over VLFM. Even using only the belief map yields a +7.7% SR and +0.69% SPL gain, showing that 3D voxel-based belief map effectively captures spatial structure and supports downstream navigation.
>
> > **[W2]:** Placing substantial details in the supplementary material may hinder accessibility for non-expert readers; including more key content in the main paper would improve clarity.
>
> **Answer:**
> Thank you for the suggestion. Due to system complexity and space limits, detailed components are in the appendix, while the main paper focuses on the most essential aspects.
>
> > **[W3]:** Although the system achieves impressive results, some design choices in the paper, such as the posterior fusion strategy and the various scorers, appear somewhat heuristic or empirically driven, lacking sufficient theoretical justification.
>
> **Answer**
>
> **For the posterior fusion strategy**, We drew inspiration from prior work on object observation confidence [1][2] and human vision [3] to design a visibility-based posterior fusion strategy. Though empirically driven, it is grounded in prior studies and consistently improves observation accuracy and exploration efficiency in our experiments, validating its effectiveness and plausibility.
>
> **For various scorers:** The scorers, though empirically designed for indoor environments, follow a clear global-to-local spatial hierarchy that aligns image semantics with spatial scales. They are not unique—like reward functions in RL, they are flexible, interpretable, and easily adaptable to different settings (e.g., indoor vs. outdoor) without retraining, enhancing the system’s scalability.
>
> **The entire system is built upon a theoretical formulation of the object navigation problem.**
> Early in our research, we developed a detailed problem formulation (omitted due to space), which guided our system design and led to SOTA results across multiple datasets. Below is the formulation and corresponding system-level optimizations:
>
> **Object Navigation Exploration Problem formulation:**
>
> Object navigation seeks to detect a target object with minimal uncertainty and observation cost, despite limited observations and incomplete environment knowledge. We define the true object distribution as $p_d(q \mid o, \mathsf{E})$, the probability of detecting object $o$ from viewpoint $q$ in scene $\mathsf{E}$, where $q \in \mathsf{S}$, the set of all reachable viewpoints. A policy $\pi_{\omega}$, parameterized by $\omega$, selects the next viewpoint based on current location $q_t$, observed viewpoints $\mathsf{S}_t$, history $\mathsf{O}_t$, and target $o$. The goal is to minimize the expected path length from initial viewpoint $q_0$ to final viewpoint $q_e$ (where the object is detected), under $p_d$ :
>
> $$
> \\min\_{\\omega} \\; \\mathbb{E}\_{q\_e \\sim p\_d(q \\mid o, \\mathsf{E}),\\; q\_0 \\sim \\mathrm{Uniform}(\\mathsf{S})} \\; L\\left(q\_e, q\_0, \\pi\_{\\omega}(o, \\mathsf{O}\_t, q\_t, \\mathsf{S}\_t)\\right)
> $$
>
> $$
> L\\left(q\_e, q\_0, \\pi\_{\\omega}(o, \\mathsf{O}\_t, q\_t, \\mathsf{S}\_t)\\right)
> = \\sum_{t = 0}^{T-1} \\| q_{t+1} - q_t \\|
> $$
> Subject to:
>
> $$q_{t+1} = \pi_{\omega}(o, \mathsf{O}_t, q_t, \mathsf{S}_t)$$
>
> $$q_T = q_e$$
>
> Here, $L$ is the total path length; $\mathsf{O}_t$ encodes accumulated observations till time $t$; $q_T$ is the final detection viewpoint; and the belief map $\hat{p}_d^{\theta}$.
>
> Since the true distribution is intractable, we estimate it with a model $\hat{p}_d^{\theta}(q \mid o, \mathsf{O}_t)$ parameterized by $\theta$, and optimize the expected search path length based on this estimated distribution.
>
> $$
> \\min\_{\\omega} \\; \\mathbb{E}\_{q\_e \\sim \\hat{p}\_d^{\theta}(q \\mid o, \\mathsf{O_t}),\\; q\_0 \\sim \\mathrm{Uniform}(\\mathsf{S})} \\; L(q_e, q_0, \pi_{\omega}(\hat{p}_d^{\theta}(q \mid o, \mathsf{O}_t),q_t, \mathsf{S}_t)) \tag{1}
> $$
>
> $$
> L\\left(q\_e, q\_0, \\pi\_{\\omega}(\hat{p}\_d^{\theta}(q \mid o, \mathsf{O}\_t), q\_t, \\mathsf{S}\_t)\\right)
> = \\sum_{t = 0}^{T-1} \\| q_{t+1} - q_t \\|
> $$
>
> subject to:
> $$
> q\_{k+1} = \pi_{\omega}(\hat{p}\_d^{\theta}(q \mid o, O_t),q\_k, S\_k)
> $$
> $$
> k = 0, 1, \dots, T-1
> $$
> $$
> q\_T = q\_e
> $$
>
> Since the policy is optimized under the estimated distribution $\hat{p}_d^{\theta}(q \mid o, \mathsf{O}_t)$, the gap $\Delta L_t$ between expected cost between estimated and true distributions at planning step $t$  can be bounded using **Hölder’s** and **Pinsker’s** inequalities:
>
> $$
> \begin{aligned}
> | \Delta L\_t | = \bigg|  \\mathbb{E}\_{q\_e \\sim p\_d(q \\mid o, \\mathsf{E}),\\; q\_0 \\sim \\mathrm{Uniform}(\\mathsf{S})} \\; L\\left(q\_e, q\_0, \\pi\_{\\omega}(o, \\mathsf{O}\_t, q\_t, \\mathsf{S}\_t)\\right) - \\mathbb{E}\_{q\_e \\sim \\hat{p}\_d^{\theta}(q \\mid o, \\mathsf{O_t}),\\; q\_0 \\sim \\mathrm{Uniform}(\\mathsf{S})} \\; L(q_e, q_0, \pi_{\omega}(\hat{p}_d^{\theta}(q \mid o, \mathsf{O}_t),q_t, \mathsf{S}_t))  \bigg|
> \end{aligned}
> $$
>
>
> $$
> = \frac{1}{N} \sum\_{q\_0 \in S} \sum_{q\_e \in S}
> \left|  \left( \hat{p}\_d^{\theta}(q \mid o, O\_t) - p\_d(q \mid o, E) \right) \right|
> L(q\_e, q\_0, \pi\_{\omega}(\hat{p}\_d^{\theta}(q \mid o, \mathsf{O}\_t), q\_t, \mathsf{S}\_t))
> $$
>
> $$
> \leq \frac{2}{N} \sup\_{q\_e, q\_0} L(q\_e, q\_0, \pi\_{\omega}(\hat{p}\_d^{\theta}(q \mid o, \mathsf{O}\_t), q\_t, \mathsf{S}\_t)))
> \cdot | \hat{p}\_d^{\theta}(q \mid o, O\_t) - p\_d(q \mid o, E) | \_{\text{TV}}
> $$
>
> $$
> \leq \frac{1}{N} \sup\_{q\_e, q\_0} L(q\_e, q\_0, \pi\_{\omega}(\hat{p}\_d^{\theta}(q \mid o, \mathsf{O}\_t), q\_t, \mathsf{S}\_t)))
> \cdot \sqrt{2 D\_{\mathrm{KL}} \left( p\_d(q \mid o, E) \,\|\, \hat{p}\_d^{\theta}(q \mid o, O\_t) \right)}
> $$
>
> where $\sup$ denotes the maximum of $L(q_e, q_0, \pi_{\omega}(\hat{p}d^{\theta}(q \mid o, \mathsf{O}t), q_t, \mathsf{S}t))$, $|\cdot|{\mathrm{TV}}$ is the total variation distance, and $D\_{\mathrm{KL}}$ is the KL divergence. As $D\_{\mathrm{KL}}(p_d ,|, \hat{p}_d^{\theta})$ decreases, the error $| \Delta L_t |$ between true and estimated cost shrinks. Therefore, **minimizing KL divergence is essential**, it ensures that planning under the estimated distribution $\hat{p}_d^{\theta}$ yields trajectories close to the true optimal, directly improving policy performance under $p_d$.
>
> **In conclusion, the optimization process is divided into two aspect at every step:**
> 1. **Minimize KL divergence** between the estimated distribution $\hat{p}_d^{\theta}(q \mid o, O_t)$ and the true distribution $p_d(q \mid o, \mathsf{E})$ over $t = 0$ to $T$, ensuring alignment between estimated and true objectives.
> 2. **Minimize expected path length** from $q_0$ to $q_e$ (where the object is detected), using the estimated distribution $\hat{p}_d^{\theta}(q \mid o, O_t)$ as defined in Equation (1).
>
> **The relationship between BeliefMapNav and the aforementioned optimization problem**
>
> * **Estimated distribution $\hat{p}_d^{\theta}(q \mid o, O_t)$ in BeliefmapNav**:
>     *  The estimated distribution (observation belief in Sec. 3.4.2) reflects the likelihood of detecting the target object. To closely approximate the true distribution $p\_d$, **BeliefMapNav introduces three dedicated refinement modules**, aiming for high accuracy at every step.
>         * **Belief Map**: A 3D prior estimating likely target locations by combining spatial-image–aligned 3D semantics with LLM-based spatial reasoning, yielding a structured and accurate scene-level representation.
>         * **Posterior Belief Map (via Visibility Map)**: Built on the prior belief map, the posterior belief integrates real-time observation likelihoods (visibility map) to continuously refine target distribution using accumulated visual evidence. This tight coupling ensures higher accuracy and significantly reduces redundant exploration.
>         * **Ray Casting–based Observation Belief:** The observation belief ($\hat{p}_d^{\theta}(q \mid o, O_t)$) is computed via ray casting from each camera pose to simulate visibility and model occlusions. It estimates the likelihood of observing the target object at different viewpoints based on the posterior belief map, enabling accurate, view-aware refinement of the observation belief.
>
> * **Expected path length in BeliefmapNav**: Observation belief-based planning (Sec. 3.5) minimizes the expected path length in Eq. (1) by planning over the estimated observation belief, enabling efficient and goal-directed object search.
>
> **This approach, based on optimization design, ultimately leads to SOTA performance.**
>
> [1]Yokoyama, Naoki, et al. "Vlfm: Vision-language frontier maps for zero-shot semantic navigation."
>
> [2] Ginting, Muhammad Fadhil, et al. "Semantic belief behavior graph: Enabling autonomous robot inspection in unknown environments."
>
> [3] Hoppe, David, and Constantin A. Rothkopf. "Multi-step planning of eye movements in visual search.

---

> > ### Comment · Reviewer_cjFY · 2025-08-07
> >
> > Thanks for the rebuttal. I appreciate the detailed design considerations presented by the authors. However, given the highly engineering-oriented nature of the Object Goal Navigation task, I will maintain my original borderline accept score.

---

> ### Author Response · Authors · 2025-08-05
> **Reviewer Discussion Invitation and Summary of Key Responses**
>
> Dear Area Chairs and Reviewers,
>
> We would first like to thank all the reviewers for their thoughtful feedback sincerely — your comments are extremely valuable to us. We have carefully addressed every concern raised through detailed clarifications and additional experiments.
>
> **We are encouraged to see that Reviewer yGEQ felt our clarifications resolved their concerns** and chose to raise their score. **As the reviewer–author discussion period approaches its deadline (August 6th)**, we warmly invite more reviewers to engage in the discussion, which would greatly help us further improve the quality of our work. If our responses have helped address your concerns, we would be truly grateful if you would consider updating your score accordingly.
>
> Below, we provide a summary of our responses to each reviewer’s specific concerns:
>
> **Reviewer kq75**:
> 1. We clarified the differences and advantages of our approach over VLFM from multiple perspectives, achieving significantly better performance under the same pipeline **(SR: +20.2%, SPL: +8.6%).**
> 2. Compared to SG-Nav and UniGoal, our method consistently outperforms on HM3D **(SR: +12.6%, SPL: +21.9%)**. On MP3D, while SR is slightly lower **(–9%)**, we still achieve a notable gain in SPL **(+7.3%)**. The detailed analysis is provided in the response; **the lower performance is likely due to dataset characteristics rather than limitations of our method.**
> 3. We provide evidence from multiple aspects showing that integrating only the top three high-confidence patches into the voxel map does **not** lead to unstable mapping or limited semantic generalization. Furthermore, experiments show that adding more patches does **not** yield meaningful performance gains.
>
> **Reviewer RDXG:**
> 1. Coincidentally, both SG-Nav and UniGoal were mentioned by different reviewers, and we have included comparisons with both in our response.
> 2. We clarify how the proposed method improves navigation efficiency from the perspectives of global optimization, real-time adaptive planning, and empirical results **(SR: +11.6%, SPL: +7.85%).**
> 3. Additionally, the paper **includes ablation studies using alternative planners.**
>
> **Reviewer cjFY:**
> 1. We provide theoretical support for our system design through **optimization modeling** and **problem decomposition** of the object navigation task.
> 2. We added a set of experiments to demonstrate the importance of the **planner** and **belief map**, as well as their contributions to performance improvement **(belief map: SR +7.7%, SPL +0.69%; planner + belief map: SR +20.2%, SPL +8.6%).**
>
> **Reviewer yGEQ:**
> 1. We analyze the limitations of directly using VLMs for object navigation from multiple perspectives, and highlight the advantages of incorporating a voxel map.
> 2. We also explain the core reason behind the observed improvement in SPL.
>
> Your feedback is highly appreciated and plays a crucial role in shaping our work. We remain fully open to addressing any remaining questions and welcome further discussion.
>
> Sincerely,
> Authors

---

### Official Review · Reviewer_kq75 · 2025-07-04

**Clarity:** 3
**Significance:** 2
**Originality:** 2
**Rating:** 3
**Confidence:** 4

**Summary:**

This paper proposes BeliefMapNav for zero-shot object navigation.
It introduces a 3D voxel-based belief map that integrates hierarchical semantic priors (scene, region, object levels) with LLM-generated landmark guidance.
The system leverages multi-scale CLIP features, fuses hierarchical semantics into a dense voxel grid, and uses a belief-driven planning module to guide exploration efficiently.
Experimental results show performance on MP3D and HM3D benchmarks.

**Questions:**

- While voxel belief maps are expressive, do they introduce storage or scalability issues in large-scale or urban-scale indoor environments?
- Several hyperparameters (e.g., fusion weights of semantic scorers, patch selection thresholds) appear to affect performance. Has the robustness under varying settings been evaluated?
- Only the top-confidence patch is used for voxel update. Have the authors considered aggregating multiple high-confidence patches to improve semantic completeness?
- In the multi-scale feature scoring module, how are patch scores across different scales aligned or normalized? Could inconsistent scoring across scales affect reliability?

**Ethical Concerns:**

["NO or VERY MINOR ethics concerns only"]

**Final Justification:**

After the rebuttal, I believe my concerns regarding the incremental nature of the technical contributions and the unresolved issues on MP3D and real-world performance have not been adequately addressed. I will maintain my rating for borderline reject.

**Limitations:**

Yes

**Quality:**

3

**Strengths And Weaknesses:**

## Strengths
- The paper proposes a structured voxel-based prior belief map that fuses three levels of semantic abstraction, supporting spatial reasoning across scenes.
- The modular design is clear and extensible.

## Weakness
- Limited novelty: While the proposed voxel-based belief map structure enhances spatial expressiveness, the overall navigation pipeline remains within the conventional frontier-based planning paradigm. There is no substantial innovation in goal reasoning or exploration strategy compared to prior works such as VLFM.
- The paper does not include comparisons with recent representative methods such as SG-Nav and Uni-Goal, which limits clarity regarding its advantages in semantic mapping and generalization.
- Each frame projects at most three high-confidence patches into the voxel map, which may lead to unstable mapping or limited semantic generalization, especially in occluded or large environments.
- Lack of real-world validation:

---

> ### Author Rebuttal · Authors · 2025-07-28
>
> > **[W1]:** Limited novelty: The overall navigation pipeline remains within the conventional frontier-based planning paradigm. There is no substantial innovation in goal reasoning or exploration strategy compared to prior works such as VLFM.
>
>  **Answer:**
>
> Our method is **fundamentally different from and outperforms prior works such as VLFM [1] in several aspects,** and it is **not restricted to the frontier-based planning paradigm.** Although our pipeline builds upon the VLFM framework, we only introduce two components—**belief maps** and **belief map-based path planning**—yet still achieve significantly better performance on HM3D, with improvements of **SR: +20.2% and SPL: +8.6%** as shown in Table 1 on HM3D.
>
> 1. **BeliefMapNav supports sampling all spatial points for global path optimization**, not just frontiers. Frontiers are used merely to reduce computation, as they tend to have higher detection probabilities. **This ensures efficiency without restricting the planner to frontier-based strategies.**
>
> 2. Our method **outperforms VLFM** and **adopts a fundamentally different approach in following aspects**.
>
>     * **In Spatial Semantic Representation**:
>         * **3D semantic Representation:** Unlike VLFM[1] and other 2D value map-based methods that fail to capture semantics across height, the 3D belief map naturally **supports vertical reasoning**, accurately models relations like “a vase on a table,” and **enables robust multi-floor navigation.**
>         * **Semantic scale alignment via spatial hierarchy:** Prior methods[1,4] project multi-scale semantics to BEV, but mismatches between image and real-world scales lead to inconsistency and blurring. BeliefMapNav addresses this using a scene–region–object hierarchy and a confidence-based scorer to remove cross-scale noise, ensuring spatially consistent semantics and accurate image-to-space mapping.
>
>     * **In Target Object Location Estimation**
>         *  **Dynamic belief estimation with semantic and observation feedback:** Fuses 0.25m voxel-level priors—based on spatial reasoning—with observation likelihoods to accurately estimate target locations at centimeter-level precision, reducing misjudgments and redundant exploration.  In contrast, methods like VLFM lack such real-time refinement for efficient navigation.
>         * **Precise observation belief estimation with Line-of-Sight Awareness:** The 3D voxel map accurately estimates the belief of observing an object from different camera poses, accounting for occlusion and line-of-sight relationships. Other methods, such as VLFM, do not account for this crucial factor.
>     * **In Efficient Exploration Strategy**
>         * Unlike VLFM’s greedy frontier selection, BeliefMapNav globally optimizes exploration paths by jointly considering all frontiers and target location probabilities. This enables real-time replanning with broad early exploration and fine-grained refinement later, reducing backtracking and improving efficiency. As shown in Table 1, the planner brings **SR ↑11.6% and SPL ↑7.85%.**
>
> **table 1: Ablation study of belief map and path planning on HM3D**
>
> |Method|SR↑|SPL↑|
> |-|-|-|
> |VLFM[1]|52.0|29.1|
> |VLFM+Belief Map|56.0|29.3|
> |BeliefMapNav(VLFM + Belief Map + Path Planning)|**62.5**|**31.6**|
>
> > **[W2]:** The paper does not include comparisons with recent representative methods such as SG-Nav and Uni-Goal.
>
> **Answer:**
>
> We identified these two excellent methods early on, but as both rely on scene graph representations, we excluded them from direct comparison. If there is a chance, **I will definitely include them in the experimental table.**  As shown in Table 2, our **approach still demonstrates clear performance gains over them**.
>
> **Table2: comparisons with SG-Nav and UniGoal**
> |Method|SR↑(HM3D)|SPL↑(HM3D)|SR↑(MP3D)|SPL↑(MP3D)|
> |-|-|-|-|-|
> | SG-Nav[2]|54.0| 24.9| 40.2|16.0|
> | UniGoal[3]|54.5|25.1|**41.0**|16.4|
> | BeliefmapNav|**61.4**| **30.6**|37.3|**17.6**|
>
> **The experimental data analysis and Limited Improvement on MP3D analysis are in response of W2 to reviewer RDXG.**
>
> > **[w3]:** Each frame projects at most three high-confidence patches into the voxel map, which may lead to unstable mapping or limited semantic generalization, especially in occluded or large environments.
>
> **Answer:**
>
> Our method is explicitly designed to avoid issues such as unstable mapping or limited semantic generalization. Below, we provide the rationale and supporting experimental evidence:
>
> * The three high-confidence patches are **spatially aligned and adaptively selected from multi-scale features**, ensuring rich semantic content. As shown in Table 3, increasing the image scale beyond 3 minimal SR improvement (62.5 → 62.8) but reduces SPL (31.6 → 29.8) due to noisy fine-grained details.
> * **The CLIP embedding itself contains rich and sufficiently general semantic information.** We use smaller-scale inputs to provide finer details, so there is no issue of limited semantic generalization.
> *  We use a **fixed-size embedding codebook,** with the semantic map storing only keys. When full, new embeddings are merged with similar ones to enable **semantic generalization and memory-efficient multi-frame fusion** in occluded or large scenes. **As this is not our main contribution, it was omitted from the paper.**  Integration works as follows:
> 	1.	**Cosine Similarity Check:** For each new embedding, cosine similarity is computed against all existing ones; if the maximum exceeds a threshold, the closest embedding is updated by new embedding the via weighted average.
> 	2.	**Handling Novel Embeddings:** If a new embedding is below the similarity threshold with all existing ones, the function finds the two most similar codebook entries and replaces the one with higher similarity to the new embedding.
>
> Table 3: Ablation study of Image Scales on HM3D
> |Image Scales(K)|SR(HM3D)|SPL(HM3D)|
> |-|-|-|
> |1|57.8|29.1|
> |2|61.3|29.3|
> |3|62.5|**31.6**|
> |4|**62.8**|29.8|
>
> > **[w4]:** Lack of real-world validation
>
> **Answer :**
>
> We are validating on a humanoid robot, **focusing on the 3D voxel-based belief map’s ability to accurately locate target objects for fine-grained manipulation, even without direct observation. It runs smoothly on Jetson Orin NX 16G at 2.78 FPS, and we expect to share the demo very soon.**
>
> We validate BeliefMapNav on Jetson from two aspects:
> * computation resources:
>     * VRAM+RAM:**8.43G**
> * computation efficiency:
>     * 3D hierarchical semantic map update:  **average 0.141s**
>     * posterior belief map + path planning: **average 0.873s**
>     The semantic map updates at every step, while the posterior belief map and planner are triggered every 4 steps to compute the next goal. BeliefmapNav runs at **2.78 FPS** on real hardware, which is acceptable for static environments.
>
> > **[q1]:** While voxel belief maps are expressive, do they introduce storage or scalability issues in large-scale or urban-scale indoor environments?
> >
> **Answer :**
>
> **We don’t have storage or scalability issues**. As mentioned in the W3 response, we use a **fixed-size embedding codebook**, which allows efficient embedding fusion and storage in large environments without memory overload or scalability problems.
>
> > **[q2]:** Several hyperparameters appear to affect performance. Has the robustness under varying settings been evaluated?
> >
> **Answer :**
>
> We did design and tune some hyperparameters, **but they are highly robust across scenes**:
>
> **Fusion weights of semantic scorers** were carefully tuned to align with spatial scales, but they remain robust as long as two principles are followed:
> 1. **Within each scorer:** Different physical units yield values of varying magnitudes, so we scale terms to make them comparable.
> 2. **Across scorers:** Scorer outputs differ in range; applying appropriate weights brings them to a similar scale.
>
> But, **the tuned weights are robust across scenes**: The scorers are based on 3D spatial quantities, which are invariant to viewpoint and unaffected by partial observations—ensuring consistency across environments.
> * **For the patch selection** in our method keeps only the top-scoring patches based on the scorer, without using an explicit threshold.
>
> > **[q3]:** Only the top-confidence patch is used for voxel update. Have the authors considered aggregating multiple high-confidence patches to improve semantic completeness?
> >
> **Answer :**
>
> Thank you for your suggestion. We did consider this approach, but the key challenge is **defining high confidence,** which is hard to generalize across scenes due to varying score ranges from different scorers in different scenes.  Instead, we leverage the stable relative magnitudes of 3D spatial metrics and apply a max operation to select top-scoring patches. This **ensures only strong signals are retained**, and **preventing low-confidence features from affecting the semantic representation.**
>
> So we  **integrate semantics in the semantic embedding space.** As noted in our **W3 response**, we use a **fixed-size embedding codebook**, enabling efficient fusion of similar semantics. **This approach improves semantic completeness while remaining memory-efficient.**
>
> > **[q4]:** In the multi-scale feature scoring module, how are patch scores across different scales aligned or normalized? Could inconsistent scoring across scales affect reliability?
>
> **Answer:**
>
> * All patches are resized to a unified resolution across image scales, ensuring consistent feature distribution.
> * Spatial attributes—such as volume, density, and instance count—remain invariant to image scale.
> * Thus, no inconsistency in scoring occurs across spatial scales.
>
> [1] "Vlfm: Vision-language frontier maps for zero-shot semantic navigation."
>
> [2] "Sg-nav: Online 3d scene graph prompting for llm-based zero-shot object navigation."
>
> [3] "Unigoal: Towards universal zero-shot goal-oriented navigation."
>
> [4] Gamap: Zero-shot object goal navigation with multi-scale geometric-affordance guidance.

---

> > ### Comment · Reviewer_kq75 · 2025-08-08
> > **Post-Rebuttal Comments**
> >
> > I appreciate the authors’ detailed rebuttal, which has addressed some of my earlier questions. However, I still have the following concerns:
> >
> > - For W1:
> >
> > (1) The proposed “multi-scale semantics” essentially adds more semantic prompts (scene–region–object), which appears conceptually similar to the approach in GaMap.
> >
> > (2) Regarding “Target Object Location Estimation,” compared to VLFM—which projects the computed text–visual similarity onto the map via FOV—the proposed method projects it via 3D voxels. While there is a technical difference from prior work, I feel this distinction is incremental and does not introduce a particularly novel insight.
> >
> > - For W2:
> >
> > In the rebuttal, the authors provided comparisons with SG-Nav and UniGoal on HM3D, but not on MP3D. In fact, on MP3D, the proposed method performs notably worse than these baselines: for example, UniGoal achieves 41.0 SR and 16.4 SPL, whereas the proposed method reports 37.3 SR and 17.6 SPL—showing a gap of nearly 4% in SR. Furthermore, for the HSSD dataset, the current SOTA has not been evaluated, which makes it difficult to contextualize the results. These points raise questions about the robustness of the method.
> >
> > - For W4:
> >
> > The rebuttal states that the proposed method can feasibly run on ARM architectures. However, this does not fully demonstrate its performance in realistic environments—especially given that some robotic platforms run on x86 architectures. My original W4 concern was an extension of W2’s robustness question, and I do not feel the rebuttal adequately addressed it.
> >
> >
> > Overall, while the rebuttal clarified certain aspects, my primary concerns remain insufficiently resolved.

---

> ### Author Response · Authors · 2025-08-05
> **Reviewer Discussion Invitation and Summary of Key Responses**
>
> Dear Area Chairs and Reviewers,
>
> We would first like to thank all the reviewers for their thoughtful feedback sincerely — your comments are extremely valuable to us. We have carefully addressed every concern raised through detailed clarifications and additional experiments.
>
> **We are encouraged to see that Reviewer yGEQ felt our clarifications resolved their concerns** and chose to raise their score. **As the reviewer–author discussion period approaches its deadline (August 6th)**, we warmly invite more reviewers to engage in the discussion, which would greatly help us further improve the quality of our work. If our responses have helped address your concerns, we would be truly grateful if you would consider updating your score accordingly.
>
> Below, we provide a summary of our responses to each reviewer’s specific concerns:
>
> **Reviewer kq75**:
> 1. We clarified the differences and advantages of our approach over VLFM from multiple perspectives, achieving significantly better performance under the same pipeline **(SR: +20.2%, SPL: +8.6%).**
> 2. Compared to SG-Nav and UniGoal, our method consistently outperforms on HM3D **(SR: +12.6%, SPL: +21.9%)**. On MP3D, while SR is slightly lower **(–9%)**, we still achieve a notable gain in SPL **(+7.3%)**. The detailed analysis is provided in the response; **the lower performance is likely due to dataset characteristics rather than limitations of our method.**
> 3. We provide evidence from multiple aspects showing that integrating only the top three high-confidence patches into the voxel map does **not** lead to unstable mapping or limited semantic generalization. Furthermore, experiments show that adding more patches does **not** yield meaningful performance gains.
>
> **Reviewer RDXG:**
> 1. Coincidentally, both SG-Nav and UniGoal were mentioned by different reviewers, and we have included comparisons with both in our response.
> 2. We clarify how the proposed method improves navigation efficiency from the perspectives of global optimization, real-time adaptive planning, and empirical results **(SR: +11.6%, SPL: +7.85%).**
> 3. Additionally, the paper **includes ablation studies using alternative planners.**
>
> **Reviewer cjFY:**
> 1. We provide theoretical support for our system design through **optimization modeling** and **problem decomposition** of the object navigation task.
> 2. We added a set of experiments to demonstrate the importance of the **planner** and **belief map**, as well as their contributions to performance improvement **(belief map: SR +7.7%, SPL +0.69%; planner + belief map: SR +20.2%, SPL +8.6%).**
>
> **Reviewer yGEQ:**
> 1. We analyze the limitations of directly using VLMs for object navigation from multiple perspectives, and highlight the advantages of incorporating a voxel map.
> 2. We also explain the core reason behind the observed improvement in SPL.
>
> Your feedback is highly appreciated and plays a crucial role in shaping our work. We remain fully open to addressing any remaining questions and welcome further discussion.
>
> Sincerely,
> Authors

---

> ### Author Response · Authors · 2025-08-07
>
> Dear Reviewer kq75,
>
> First of all, thank you again for the time and effort you have dedicated to reviewing our work — we truly appreciate your thoughtful feedback. Your comments are extremely important to us and play a vital role in helping improve the quality of our research. **As the reviewer–author discussion period approaches its deadline, I sincerely hope there is still enough time for me to clarify our responses and address any remaining concerns you might have.** Our responses **have already helped resolve the concerns raised by the other two reviewers,** which also resulted in improved scores. If any part of our response was unclear or if you still have questions, we would be more than happy to provide further clarification and engage in additional discussion.
>
> **Additionally: The experimental analysis details of the comparison between BeliefMapNav, SG-Nav, and UniGoal on HM3D and MP3D from the response of Reviewer RDXG**
>
> **Table 1: comparisons with SG-Nav and UniGoal**
> | Method       | SR↑(HM3D) | SPL↑(HM3D) | SR↑(MP3D) | SPL↑(MP3D) |
> |--------------|-----------|------------|-----------|------------|
> | SG-Nav       | 54.0      | 24.9       | 40.2      | 16.0       |
> | UniGoal     | 54.5      | 25.1       | **41.0**      | 16.4       |
> | BeliefmapNav | **61.4**      | **30.6**       | 37.3      | **17.6**       |
>
> We identified these two strong methods early on, but as both rely on scene graph representations, we excluded them from direct comparison. Nonetheless, As shown in table 2 our **approach still demonstrates clear performance gains over them**.
>
> * **Experimental data analysis:**
> As shown in the table, BeliefMapNav significantly outperforms prior **SOTA methods in both SR and SPL on HM3D**, highlighting its strong capability in complex and high-quality environments. While its SR on MP3D is slightly lower, it **still achieves a notably higher SPL**, demonstrating more efficient and purposeful navigation compared to other methods.
>     *	On HM3D, BeliefMapNav achieves 61.4% SR and 30.6% SPL, **significantly surpassing SG-Nav and UniGoal** by a large margin in both success rate and path efficiency(SPL). This demonstrates the strength of our approach in accurately reasoning over spatial structure and planning globally optimal paths.
>     *	On MP3D, while BeliefMapNav’s SR is slightly lower, it still yields the highest SPL (17.6) among all methods, indicating that it navigates with the most efficient and purposeful trajectories, even under more noisy or degraded conditions.
>
> * **Limited Improvement on MP3D**
> The slightly lower SR of BeliefMapNav on MP3D can be **attributed to two dataset-specific limitations, rather than a weakness of the method itself:**
>
>     * **Low mesh quality hinders object recognition:** MP3D’s poor mesh makes targets hard to detect. SG-Nav mitigates this with re-perception, slightly improving SR. BeliefMapNav instead localizes targets via belief map–guided planning, and despite not enhancing detection, still leads in SPL through optimized exploration.
>
>     * **Broken surfaces and holes distort observation beliefs**: MP3D’s mesh flaws disrupt BeliefMapNav’s ray tracing, causing false high beliefs in non-traversable but visually reachable areas. This leads to some inefficiency and slight SPL drop, yet BeliefMapNav still achieves strong SPL gains, demonstrating robustness to noise.
>
>
> Thank you again for your valuable time and consideration.
>
> Sincerely, Authors

---

> ### Author Response · Authors · 2025-08-08
>
> Thank you again for your thoughtful question and suggestion. We apologize for the oversight and for not addressing these points more clearly in our initial response. However, we are able to provide clear and satisfactory answers to these questions.
>
> > **For W1:** The proposed “multi-scale semantics” essentially adds more semantic prompts (scene–region–object), which appear conceptually similar to the approach in GaMap.
>
> **Answer:**
> While both methods adopt “multi-scale semantic”, they are **fundamentally different** in nature.
> 1. **Image VS spatial multi-scale**: **GaMap applies multi-scale semantics only at the image level, without aligning to actual 3D spatial scales.**  This misalignment leads to inconsistent and misleading value estimation, especially when objects appear at varying depths.
> 2. **Max value updating VS Feature fusion**: GAMap update based on image-text similarity, discarding spatial semantics. This makes them brittle—max-based updates **amplify noisy observations from depth changes.** In contrast, BeliefMapNav performs target-agnostic fusion of multi-scale spatial semantic embeddings—**automatically selecting appropriate spatial scales—resulting in a more robust and consistent semantic map.**
>
> **These differences lead to a significant performance gap between the two methods in object navigation:**
>
> 1. **GAmap: Persistent false-positive trap:**
>
>     GaMap updates a **2D value map** using max-pooling **without multi-scale spatial semantics.** This **leads to persistent false positives**—e.g., when the agent is near a sofa while searching for a bed, image-level features may wrongly assign high confidence to the sofa region. **Without spatial alignment, both large and small image-scale features reinforce this error,** making the value always high.
>
>     BeliefMapNav uses a 3D hierarchical semantic voxel map that fuses multi-scale spatial semantic features according to their spatial-scale relationships, and updates independently of the target object, **preventing early false high value from being reinforced over time.** By incorporating spatial multi-scale semantics—including high-level scene context like “living room”—**it can downweight regions unlikely to contain the target (e.g., a bed is unlikely to appear in a living room)**, enabling more robust and context-aware navigation. And as observations accumulate, **spatial semantics become increasingly accurate.**
>
> 2. **GAmap: Lacks holistic spatial reasoning ability:**
>     1. **Image and spatial scale misalignment:** It projects image-scale values directly onto a 2D map without accounting for actual 3D spatial scale or depth, causing features from different depths to be incorrectly aligned in space. **This leads to semantic distortion**—for example, local details from a close view may overwrite broader scene context at the correct spatial scale.
>     2. **Value updating:** GaMap updates its value map using a max operation tied to the current target, discarding lower-confidence cues from past views. **This leads to fragmented, overfitted semantics with no accumulation of spatial semantics.**
>
> **These limitations make global reasoning difficult.** The agent **lacks a coherent spatial understanding**, increasing the **risk of local optima or inefficient search.** In contrast, BeliefMapNav **maintains target-agnostic**, spatially **aligned updates in a 3D voxel hierarchy,** enabling more consistent, robust navigation. As shown in Table 1.
>
> **More shortcomings of the value map based methods are shown in response to VLFM.**
>
> **Table 1: comparison of GAMap**
> |Methods|SR(HM3D)|SPL(HM3D)|
> |-|-|-|
> |GAMap|53.1|26.0|
> |BeliefmapNav|**61.4**|**30.6**|

---

> ### Author Response · Authors · 2025-08-08
>
> > **For W1:** Regarding “Target Object Location Estimation,” compared to VLFM—which **projects the computed text–visual similarity onto the map via FOV**—the proposed method **projects it via 3D voxels.** While there is a technical difference from prior work, I feel this distinction is incremental and does not introduce a particularly novel insight.
>
> **Answer:**
> From your concern, we realize we may not have clearly communicated the fundamental difference between BeliefMapNav and VLFM—our apologies for the confusion. Unlike VLFM, which projects text–visual similarity scores onto the map via FOV, **our method projects multi-scale spatial semantic features into 3D voxels not the belief value**. This demonstrates that our method is not an incremental improvement, **but a fundamentally different approach.**
>
> Importantly, in BeliefMapNav, **the belief score is not directly projected; it is inferred by the LLM through reasoning over the 3D hierarchical semantic voxel map**, leveraging accumulated spatial semantics rather than single-view similarity. The following are the **two most important distinctions**:
>
> 1. Unlike VLFM, which projects **target-similarity scores(Value Map)** into BEV space, BeliefMapNav projects **multi-scale, spatially aligned semantic embeddings(3D Hierarchical Semantic Voxel Map)** in to 3D space:
>     * **Spatial reasoning:** Value maps are shallow and target-specific, lacking structure or context. **Unable to reason globally**—can only update the value map based on the current observation. In contrast, semantic maps preserve multi-scale spatial cues—**enabling LLMs to reason robustly over the whole observed 3D space with spatial context and accumulated knowledge** for more robust and generalizable exploration.
>     * **Fine-Grained Localization:** Image-text similarity scores are often coarse and ambiguous, **especially in cluttered or visually similar scenes.** In contrast, BeliefMapNav’s multi-scale spatial semantic map offers higher resolution and discriminative power, **enabling more precise localization through rich, context-aware representations.**
>     * **Multi-object task:** Value maps are **tied to a single target,** requiring separate updates for each goal. In contrast, BeliefMapNav’s semantic map is **target-agnostic,** naturally supporting multi-goal queries and scalable to complex tasks.
>     * **Interpretability:** Value maps offer only target-specific scalar values, revealing little about why a region is activated. BeliefMapNav preserves rich, multi-scale semantic features—enabling spatial context reasoning and making decisions more transparent and explainable.
> 3. **VLFM fuse target-specific values, BeliefMapNav fuses spatially aligned semantic features:**
>     * **Map Robustness:** Value-based fusion (e.g., max/avg) is **sensitive to noise and accidental mismatches,** with no reliable way to filter or reverse updates. BeliefMapNav’s feature fusion **filters unreliable observations** (e.g., overly close views), retains scale-consistent features, and merges them via a semantic codebook—resulting in a more stable and robust map.
>     * **Semantic Generalization:** Value fusion aggregates isolated values **without building consistent semantics.** Feature fusion integrates evidence across views, enabling more general, robust, and view-invariant representations.
>     * **Semantic Richness:** Value-based fusion retains only a scalar value per location, discarding rich contextual information. In contrast, BeliefMapNav preserves full semantic embeddings at multiple spatial scales—capturing object, region, and scene-level semantics—enabling deeper reasoning and more informative navigation decisions.
>
> * **table 1: Ablation study of 3d voxel- based belief map and path planning**
> | Method| SR↑| SPL↑|
> |-|-|-|
> | VLFM| 52.0| 29.1|
> | VLFM+belief map| 56.0| 29.3|
> | BeliefMapNav(VLFM + belief map + path planning) | **62.5** | **31.6** |

---

> ### Author Response · Authors · 2025-08-08
>
> > **For W2:** The rebuttal includes HM3D comparisons but omits MP3D, where the method underperforms **(e.g., 37.3 SR vs. UniGoal’s 41.0).** HSSD results also lack SOTA baselines, making it hard to assess robustness.
>
> **Answer**
> We appreciate your mention of SG-Nav and UniGoal—both are excellent works. We have compared our method against them **on both MP3D and HM3D in the initial rebuttal response as shown in Table2** While our SR on MP3D is slightly lower, this is largely due to **their use of object verification to reduce FP of detector under poor mesh quality(MP3D)**. While object verification to reduce detector false positives under poor mesh quality (e.g., MP3D) is important, **it is orthogonal to our contribution, which focuses on efficient target-oriented exploration**. This is also why, on HM3D—where mesh quality is higher—our method significantly outperforms baselines in both **SR(+12.7%) and SPL(+21.9%).**
> 1. **Our main contribution lies in efficient exploration in unseen environments using belief maps**. This is clearly reflected in our **significantly better SPL scores—for efficient,** we outperform SG-Nav by **+22.9%** on HM3D and **+10%** on MP3D, and UniGoal by **+21.9%** on HM3D and **+7.3%** on MP3D—**demonstrating the effectiveness of our exploration strategy.**
>
> 2. It’s also important to note that the **mesh quality of MP3D is substantially lower than that of HM3D** In the error analysis, we observe that the detector’s **(FP) in MP3D failure cases is significantly higher (73.9%) compared to that in HM3D (44.3%).** Therefore, without employing object verification, the SR gap on MP3D is largely **due to detector limitations and poor mesh quality of MP3D, rather than shortcomings in our exploration design.**
>
> 3. Another key factor affecting SPL on MP3D is the quality of its mesh. **MP3D contains numerous broken surfaces and holes,** and since our method **explicitly models occlusions within the FOV,** these imperfections introduce noise and reduce exploration efficiency.
>
> 4. HSSD is a relatively **new benchmark** but has already attracted significant attention, and **it is officially supported by Habitat.** Its mesh quality is substantially higher than that of HM3D and MP3D, which **greatly reduces the impact of detector FP and mesh holes.** As a result, our method achieves a significant performance advantage on HSSD **(SR:+27.8%,SPL:+22.4%).**
>
> Lastly, in early experiments, SG-Nav achieved **SR: 41.7 and SPL: 20.4** on HM3D, which deviates from the results reported in the paper **(SR: 54.0, SPL: 24.9).** This discrepancy was also one of the reasons we did not include it in our comparisons.
>
> **Table 2: comparisons with SG-Nav and UniGoal,same as the initial rebuttal**
>
> |Method|SR↑(HM3D)|SPL↑(HM3D)|SR↑(MP3D)|SPL↑(MP3D)|
> |-|-|-|-|-|
> |SG-Nav|54.0|24.9|40.2|16.0|
> |UniGoal| 54.5| 25.1|**41.0**|16.4|
> |BeliefmapNav|**61.4**|**30.6**|37.3|**17.6**|
>
> > **For W4:** The rebuttal states that the proposed method can feasibly run on ARM architectures. However, this does not fully demonstrate its performance—especially some robotic platforms run on x86 architectures. My original W4 concern was an extension of W2’s robustness question, and I do not feel the rebuttal adequately addressed it.
> 1. **Jetson Compatibility:** NVIDIA’s Jetson series is **widely used in robotics.** BeliefMapNav runs smoothly on the **Unitree G1 robot** with Jetson Orin. While we’re still addressing localization error challenges specific to humanoid robots, **the BeliefmapNav is already efficient and reliable on embedded hardware.**
> 2. **X86 Evaluation:** **All experiments in our paper were conducted on standard x86 machines,** where BeliefMapNav runs efficiently and stably, demonstrating its compatibility with X86 computing environments
>
> **We look forward to your new feedback. If there are any remaining questions or aspects that require further clarification, we would greatly appreciate your feedback.**

---

> ### Author Response · Authors · 2025-08-09
>
> Dear Reviewer kq75,
>
> Thank you again for reviewing our paper and for your time and effort. **With only 8 hours left** before the reviewer–author discussion deadline, we would like to briefly summarize our key responses (details are in our earlier replies). Apologies for sending multiple short messages due to the word limit.
> 1. **Not an incremental improvement:**
>
>     VLFM and GaMap directly project text–image similarity scores into a **BEV value map**. In contrast, we build and maintain a 3D multi-scale spatial semantic voxel map(**3D feature map**), continually fusing semantic features rather than scores. The LLM then reasons in this semantic space—using commonsense and spatial context—to generate target-related beliefs. **The detailed advantages and distinctions are provided in our earlier analysis.**
>
> 2. **MP3D performance gap:**
>
>     SG-Nav and UniGoal’s SR advantage on MP3D (poor mesh quality, high detection errors) **mainly comes from their object-detection verification mechanism, which is outside our scope.** Our contribution focuses on **efficient target-oriented exploration,** reflected in SPL. On HM3D (higher mesh quality), our method shows clear advantages. **Please refer to our earlier detailed response for detailed data and experimental analysis.**
>
> 3. **Real-world deployment:**
>
>     BeliefMapNav **runs stably on the Unitree G1 robot** (Jetson Orin) and **on both ARM and x86 platforms.** We are currently addressing localization errors in humanoid robots to ensure full stability. **Please see our earlier detailed response for more information.**
>
> We truly hope to address your concerns within this limited time. Thank you again, and have a great weekend.

---

> ### Author Response · Authors · 2025-08-09
>
> Dear Reviewer,
>
> We noticed that your rating is currently not visible, and we are not sure whether our response has sufficiently addressed your concerns. If there are any remaining issues or aspects that you feel require further clarification, we would greatly appreciate your feedback.
>
> Your feedback is extremely valuable to us, and we thank you again for your time and effort in reviewing our work.

---

### Note · Authors · 2025-08-12

Dear AC and reviwers:

We are deeply grateful to the AC and reviewers. After rebuttal, two reviews turned positive; with the prior positive, a clear majority now recommends acceptance, reflecting our clarifications on yGEQ and RDXG.

We appreciate all of you for your comments highlighting the strengths of our work for summary:
* Experiments are thorough, and performance is strong.(cjFY)
* High SPL shows the benefit of belief-map–integrated planning.(RDXG)
* An interesting idea: a 3D voxel map for embodied navigation that enables scene understanding and map-based planning.(yGEQ)

Our rebuttal resolved three reviewers’ concerns. Reviewer kq75 raised late questions from an apparent misunderstanding; we clarified before the deadline, with no follow-up. Key responses follow:
* Concern: **BeliefmapNav is incremental over VLFM and GAMap.**
    * GAMap and VLFM perform **inference in image space** and **project relevance scores onto a BEV value map.**
    * BeliefmapNav **incrementally updates a spatial, multi-scale semantic 3D voxel map**, then **employs an LLM for in-voxel reasoning for belief.**
    * Conclusion: In contrast to **value-map-based** GAMap/VLFM, BeliefmapNav reasons over **a multi-scale spatial feature map** and integrates an LLM with a probabilistic planner, yielding **efficient exploration and SOTA on MP3D/HM3D/HSSD.**

* Concern: On MP3D, our SR remains lower than UniGoal and SG-Nav.
    * The MP3D SR gap reflects UniGoal/SG-Nav’s **object-verification pipelines** for poor meshes—**outside our scope.** Our focus is **efficient belief-map exploration in unseen environments,** as shown by SPL gains: **+22.9%/+10.0%** over SG-Nav and **+21.9%/+7.3%** over UniGoal (HM3D/MP3D).
    * **MP3D’s mesh quality is much lower than HM3D’s**; detector FPs are higher on MP3D (**73.9% vs. 44.3%**). Without object verification, **the MP3D SR gap mainly reflects detector/mesh limits, not our exploration design.** On HM3D, we outperform baselines: **SR +12.7%, SPL +21.9%.**

* Concern:Real-robot validation
    * We have **successfully run BeliefmapNav on a Unitree G1**; localization remains unstable and **is being addressed.**

**Planned improvements:**
1. Real-robot deployment
2. obust multi-floor traversal to showcase 3D mapping’s benefits

We are deeply grateful to the AC and reviewers. We contribute a **3D voxel belief map** and an **efficient probabilistic planner** for object navigation, improving exploration efficiency and success.


Sincerely, Authors

---

### Decision · Program_Chairs · 2025-09-17

**Decision:**

Accept (poster)

**Comment:**

The paper proposes to use 3D voxel-based belief maps to better infer the prior for the target locations. The paper received generally borderline ratings (3 Borderline Accepts and 1 Borderline Reject). The reviewers initially raised concerns such as:
- Lack of comparisons with strong baselines such as SG-Nav and Uni-Goal.
- Lack of real-world validation.
- Heuristics such as the posterior fusion strategy and the various scorers.
- Lack of clarity regarding the factors contributing to the improvements.

The authors clarified some of the issues in the rebuttal (e.g., comparisons with SG-Nav and Uni-Goal and using stronger LLM/VLM models for object navigation). The AC checked the paper, the reviews, the rebuttal and the discussions. Overall, the paper is not particularly strong, as it is conceptually similar to previous work and primarily focuses on pipeline engineering. However, the AC believes that putting these individual components together as a system provides insights for the community. Therefore, the AC concurs with the majority of the reviewers and recommends acceptance.


P.S. Mentioning relative improvements is quite misleading. The authors should change those to absolute numbers in the revision.